# Neuronal timescales are functionally dynamic and shaped by cortical microarchitecture

Richard Gao[1]*, Ruud L van den Brink[2], Thomas Pfeffer[3], Bradley Voytek[1,4,5,6]

[1]Department of Cognitive Science, University of California, San Diego, La Jolla, United States; [2]Section Computational Cognitive Neuroscience, Department of Neurophysiology and Pathophysiology, University Medical Center Hamburg-Eppendorf, Hamburg, Germany; [3]Center for Brain and Cognition, Computational Neuroscience Group, Universitat Pompeu Fabra, Barcelona, Spain; [4]Halıcıoğlu Data Science Institute, University of California, San Diego, La Jolla, United States; [5]Neurosciences Graduate Program, University of California, San Diego, La Jolla, United States; [6]Kavli Institute for Brain and Mind, University of California, San Diego, La Jolla, United States

**Abstract** Complex cognitive functions such as working memory and decision-making require information maintenance over seconds to years, from transient sensory stimuli to long-term contextual cues. While theoretical accounts predict the emergence of a corresponding hierarchy of neuronal timescales, direct electrophysiological evidence across the human cortex is lacking. Here, we infer neuronal timescales from invasive intracranial recordings. Timescales increase along the principal sensorimotor-to-association axis across the entire human cortex, and scale with single-unit timescales within macaques. Cortex-wide transcriptomic analysis shows direct alignment between timescales and expression of excitation- and inhibition-related genes, as well as genes specific to voltage-gated transmembrane ion transporters. Finally, neuronal timescales are functionally dynamic: prefrontal cortex timescales expand during working memory maintenance and predict individual performance, while cortex-wide timescales compress with aging. Thus, neuronal timescales follow cytoarchitectonic gradients across the human cortex and are relevant for cognition in both short and long terms, bridging microcircuit physiology with macroscale dynamics and behavior.

*For correspondence:
r.dg.gao@gmail.com

**Competing interests:** The authors declare that no competing interests exist.

## Introduction

Human brain regions are broadly specialized for different aspects of behavior and cognition, and the temporal dynamics of neuronal populations across the cortex are thought to be an intrinsic property (i.e., neuronal timescale) that enables the representation of information over multiple durations in a hierarchically embedded environment (*Kiebel et al., 2008*). For example, primary sensory neurons are tightly coupled to changes in the environment, firing rapidly to the onset and removal of a stimulus, and showing characteristically short intrinsic timescales (*Ogawa and Komatsu, 2010*; *Runyan et al., 2017*). In contrast, neurons in cortical association (or transmodal) regions, such as the prefrontal cortex (PFC), can sustain their activity for many seconds when a person is engaged in working memory (*Zylberberg and Strowbridge, 2017*), decision-making (*Gold and Shadlen, 2007*), and hierarchical reasoning (*Sarafyazd and Jazayeri, 2019*). This persistent activity in the absence of immediate sensory stimuli reflects longer neuronal timescales, which is thought to result from neural attractor states (*Wang, 2002*; *Wimmer et al., 2014*) shaped by N-methyl-D-aspartate receptor (NMDA)-mediated recurrent excitation and fast feedback inhibition (*Wang, 2008*;

**eLife digest** The human brain can both quickly react to a fleeting sight, like a changing traffic light, and slowly integrate complex information to form a long-term plan. To mirror these requirements, how long a neuron can be activated for – its 'timescale' – varies greatly between cells.

A range of timescales has been identified in animal brains, by measuring single neurons at a few different locations. However, a comprehensive study of this property in humans has been hindered by technical and ethical concerns. Without this knowledge, it is difficult to understand the factors that may shape different timescales, and how these can change in response to environmental demands.

To investigate this question, Gao et al. used a new computational method to analyse publicly available datasets and calculate neuronal timescales across the human brain. The data were produced using a technique called invasive electrocorticography, where electrodes placed directly on the brain record the total activity of many neurons. This allowed Gao et al. to examine the relationship between timescales and brain anatomy, gene expression, and cognition.

The analysis revealed a continuous gradient of neuronal timescales between areas that require neurons to react quickly and those relying on long-term activity. 'Under the hood', these timescales were associated with a number of biological processes, such as the activity of genes that shape the nature of the connections between neurons and the amount of proteins that let different charged particles in and out of cells. In addition, the timescales could be flexible: they could lengthen when areas specialised in working memory were actively maintaining information, or shorten with age across many areas of the brain. Ultimately, the technique and findings reported by Gao et al. could have useful applications in the clinic, using neuronal timescale to better understand brain disorders and pinpoint their underlying causes.

*Wang, 1999*), with contributions from other synaptic and cell-intrinsic properties (*Duarte and Morrison, 2019*; *Gjorgjieva et al., 2016*). How connectivity and various cellular properties combine to shape neuronal dynamics across the cortex remains an open question.

Anatomical connectivity measures based on tract tracing data, such as laminar feedforward vs. feedback projection patterns, have classically defined a hierarchical organization of the cortex (*Felleman and Van Essen, 1991*; *Hilgetag and Goulas, 2020*; *Vezoli et al., 2020*). Recent studies have also shown that variations in many microarchitectural features follow continuous and coinciding gradients along a sensory-to-association axis across the cortex, including cortical thickness, cell density, and distribution of excitatory and inhibitory neurons (*Huntenburg et al., 2018*; *Wang, 2020*). In particular, gray matter myelination (*Glasser and Van Essen, 2011*)—a noninvasive proxy of anatomical hierarchy consistent with laminar projection data—varies with the expression of genes related to microcircuit function in the human brain, such as NMDA receptor and inhibitory cell-type marker genes (*Burt et al., 2018*). Functionally, specialization of the human cortex, as well as structural and functional connectivity (*Margulies et al., 2016*), also follow similar macroscopic gradients. Moreover, in addition to the broad differentiation between sensory and association cortices, there is evidence for an even finer hierarchical organization within the frontal cortex (*Sarafyazd and Jazayeri, 2019*). For example, the anterior-most parts of the PFC are responsible for long timescale goal-planning behavior (*Badre and D'Esposito, 2009*; *Voytek et al., 2015a*), while healthy aging is associated with a shift in these gradients such that older adults become more reliant on higher-level association regions to compensate for altered lower-level cortical functioning (*Davis et al., 2008*).

Despite convergent observations of cortical gradients in structural features and cognitive specialization, there is no direct evidence for a similar gradient of neuronal timescales across the human cortex. Such a gradient of neuronal dynamics is predicted to be a natural consequence of macroscopic variations in synaptic connectivity and microarchitectural features (*Chaudhuri et al., 2015*; *Duarte et al., 2017*; *Huang and Doiron, 2017*; *Huntenburg et al., 2018*; *Wang, 2020*), and would be a primary candidate for how functional specialization emerges as a result of hierarchical temporal processing (*Kiebel et al., 2008*). Single-unit recordings in rodents and non-human primates demonstrated a hierarchy of timescales that increase, or lengthen, progressively along a posterior-to-anterior axis (*Dotson et al., 2018*; *Murray et al., 2014*; *Runyan et al., 2017*; *Wasmuht et al., 2018*),

while intracranial recordings and functional neuroimaging data collected during perceptual and cognitive tasks suggest likewise in humans (*Baldassano et al., 2017*; *Honey et al., 2012*; *Lerner et al., 2011*; *Watanabe et al., 2019*). However, these data are either sparsely sampled across the cortex or do not measure neuronal activity at the cellular and synaptic level directly, prohibiting the full construction of an electrophysiological timescale gradient across the human cortex. As a result, while whole-cortex data of transcriptomic and anatomical variations exist, we cannot take advantage of them to dissect the contributions of synaptic, cellular, and circuit connectivity in shaping fast neuronal timescales, nor ask whether regional timescales are dynamic and relevant for human cognition.

Here we combine several publicly available datasets to infer neuronal timescales from invasive human electrocorticography (ECoG) recordings and relate them to whole-cortex transcriptomic and anatomical data, as well as probe their functional relevance during behavior (*Figure 1A* for schematic of study; *Tables 1* and *2* for dataset information). Unless otherwise specified, (*neuronal*) *timescale* in the following sections refers to ECoG-derived timescales, which are more reflective of fast synaptic and transmembrane current timescales than single-unit or population spiking timescales (*Figure 1A*, left box), though we demonstrate in macaques a close correspondence between the two. In humans, neuronal timescales increase along the principal sensorimotor-to-association axis across the cortex and align with macroscopic gradients of gray matter myelination (T1w/T2w ratio) and synaptic receptor and ion channel gene expression. Finally, we find that human PFC timescales expand during working memory maintenance and predict individual performance, while cortex-wide timescales compress with aging. Thus, neuronal timescales follow cytoarchitectonic gradients across the human cortex and are relevant for cognition in both short and long terms, bridging microcircuit physiology with macroscale dynamics and behavior.

## Results

### Neuronal timescale can be inferred from the frequency domain

Neural time series often exhibit time-lagged correlation (i.e., autocorrelation), where future values are partially predictable from past values, and predictability decreases with increasing time lags. For demonstration, we simulate the aperiodic (non-rhythmic) component of ECoG recordings by convolving Poisson population spikes with exponentially decaying synaptic kernels with varying decay constant (*Figure 1B*). Empirically, the degree of self-similarity is characterized by the autocorrelation function (ACF), and 'timescale' is defined as the time constant ($\tau$) of an exponential decay function ($e^{-\frac{t}{\tau}}$) fit to the ACF, i.e., the time it takes for the autocorrelation to decrease by a factor of $e$ (*Figure 1C*).

Equivalently, we can estimate timescale in the frequency domain from the power spectral density (PSD). PSDs of neural time series often follow a Lorentzian function of the form $\frac{1}{f_k^2 + f^2}$, where power is approximately constant until the 'knee frequency' ($f_k$, *Figure 1D*), then decays following a power law. This approach is similar to the one presented in *Chaudhuri et al., 2017*, but here we further allow the power law exponent (fixed at two in the equation above) to be a free parameter representing variable scale-free activity (*He et al., 2010*; *Miller et al., 2009*; *Podvalny et al., 2015*; *Voytek et al., 2015c*). We also simultaneously parameterize oscillatory components as Gaussians peaks, allowing us to remove their effect on the power spectrum, providing more accurate estimates of the knee frequency. From the knee frequency of the aperiodic component, neural timescale (decay constant) can then be computed exactly as $\tau = \frac{1}{2\pi f_k}$.

Compared to fitting exponential decay functions in the time domain (e.g., *Murray et al., 2014*)—which can be biased even without the presence of additional components (*Zeraati et al., 2020*)—the frequency domain approach is advantageous when a variable power law exponent and strong oscillatory components are present, as is often the case for neural signals (example of real data in *Figure 1D*). While the oscillatory component can corrupt naive measurement of $\tau$ as time for the ACF to reach 1/e (*Figure 1D*, inset), it can be more easily accounted for and removed in the frequency domain as Gaussian-like peaks. This is especially important considering neural oscillations with non-stationary frequencies. For example, a broad peak in the power spectrum (e.g., ~10 Hz in bandwidth in *Figure 1D*) represents drifts in the oscillation frequency over time, which is easily accounted for with a single Gaussian, but requires multiple cosine terms to capture well in the

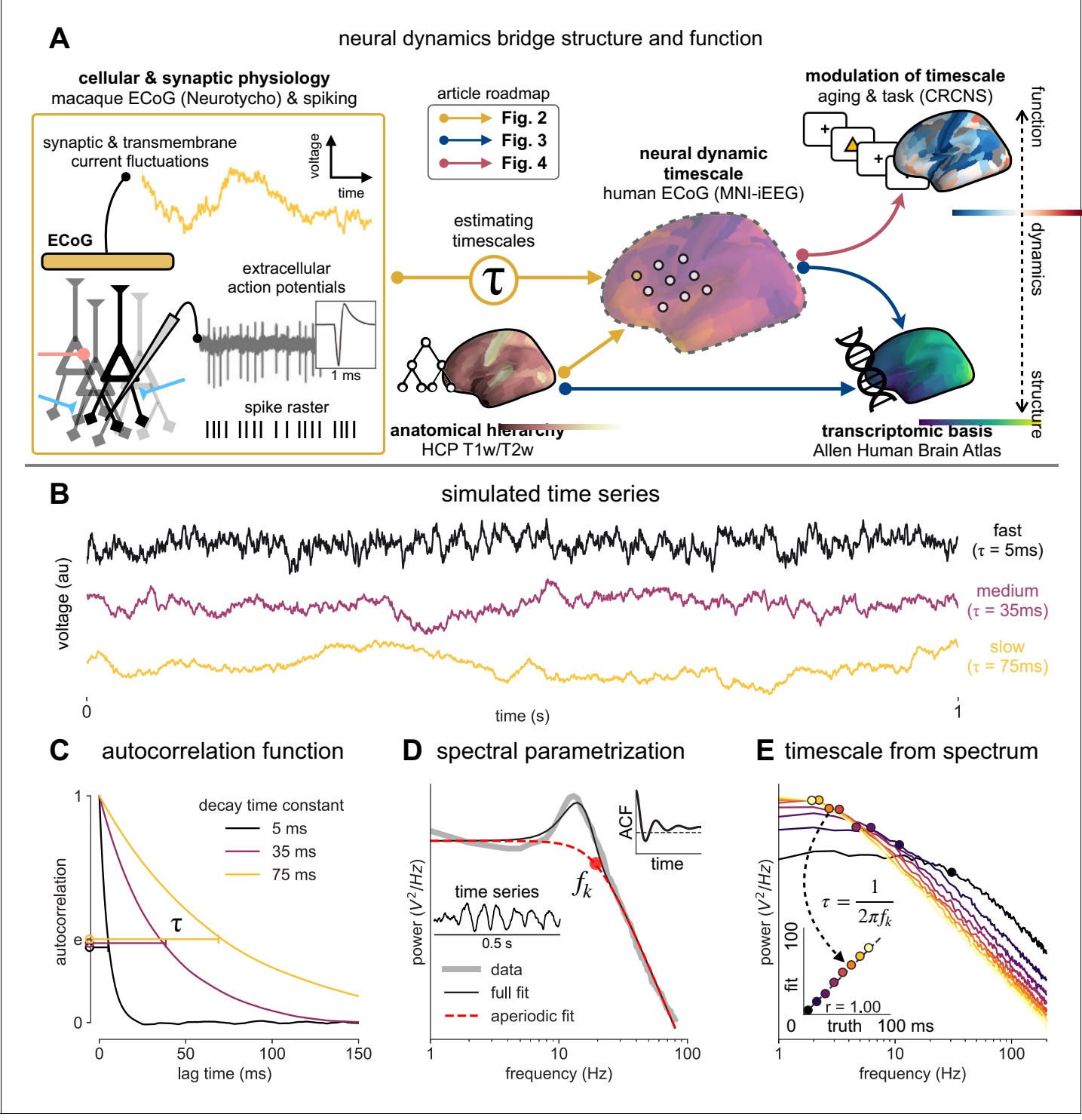

**Figure 1.** Schematic of study and timescale inference technique. (**A**) In this study, we infer neuronal timescales from intracranial field potential recordings, which reflect integrated synaptic and transmembrane current fluctuations over large neural populations (*Buzsáki et al., 2012*). Combining multiple open-access datasets (*Table 1*), we link timescales to known human anatomical hierarchy, dissect its cellular and physiological basis via transcriptomic analysis, and demonstrate its functional modulation during behavior and through aging. (**B**) Simulated time series and their (**C**) autocorrelation functions (ACFs), with increasing (longer) decay time constant, $\tau$ (which neuronal timescale is defined to be). (**D**) Example human electrocorticography (ECoG) power spectral density (PSD) showing the aperiodic component fit (red dashed), and the 'knee frequency' at which power drops off ($f_k$, red circle; insets: time series and ACF). (**E**) Estimation of timescale from PSDs of simulated time series in (**B**), where the knee frequency, $f_k$, is converted to timescale, $\tau$, via the embedded equation (inset: correlation between ground truth and estimated timescale values).

**Table 1.** Summary of open-access datasets used.

| Data | Ref. | Specific source/format used | Participant info | Relevant figures |
|---|---|---|---|---|
| MNI Open iEEG Atlas | *Frauscher et al., 2018a*; *Frauscher et al., 2018b* | | N = 105 (48 females) Ages: 13–65, 33.4 ± 10.6 | *Figure 2A–D*, *Figure 3*, *Figure 4E and F* |
| T1w/T2w and cortical thickness maps from Human Connectome Project | *Glasser et al., 2016*; *Glasser and Van Essen, 2011* | Release S1200, March 1, 2017 | N = 1096 (596 females) Age: 22–36+ (details restricted due to identifiability) | *Figure 2C and D*, *Figure 3D–F* |
| Neurotycho macaque ECoG | *Nagasaka et al., 2011*; *Yanagawa et al., 2013* | Eyes-open state from anesthesia datasets (propofol and ketamine) | Two animals (Chibi and George) four sessions each | *Figure 2E–G* |
| Macaque single-unit timescales | *Murray et al., 2014* | *Figure 1* of reference | | *Figure 2E–G* |
| Whole-cortex interpolated Allen Brain Atlas human gene expression | *Gryglewski et al., 2018*; *Hawrylycz et al., 2012* | Interpolated maps downloadable from http://www.meduniwien.ac.at /neuroimaging/mRNA.html | N = 6 (one female) Age: 24, 31, 39, 49, 55, 57 (42.5 ± 12.2) | *Figure 3* |
| Single-cell timescale-related genes | *Bomkamp et al., 2019*; *Tripathy et al., 2017* | Table S3 from *Tripathy et al., 2017*, Online Table 1 from *Bomkamp et al., 2019* | N = 170 (*Tripathy et al., 2017*) and 4168 (*Bomkamp et al., 2019*) genes | *Figure 3C and D* |
| Human working memory ECoG | *Johnson, 2019*; *Johnson, 2018c*; *Johnson et al., 2018a*, *Johnson et al., 2018b* | CRCNS fcx-2 and fcx-3 | N = 14 (five females) Age: 22–50, 30.9 ± 7.8 | *Figure 4A–D* |

autocorrelation. Therefore, in this study, we apply spectral parameterization to extract timescales from intracranial recordings (*Donoghue et al., 2020*). We validate this approach on PSDs computed from simulated neural time series and show that the extracted timescales closely match their ground-truth values (*Figure 1E*).

## Timescales follow anatomical hierarchy and are ~10 times faster than spiking timescales

Applying this technique, we infer a continuous gradient of neuronal timescales across the human cortex by analyzing a large dataset of human intracranial (ECoG) recordings of task-free brain activity (*Frauscher et al., 2018a*). The MNI-iEEG dataset contains 1 min of resting state data across 1772 channels from 106 patients (13–62 years old, 48 females) with variable coverages, recorded using

**Table 2.** Reproducing figures from code repository.

All IPython notebooks (*Gao, 2020*): **https://github.com/rdgao/field-echos/tree/master/notebooks**

| Notebook | Results |
|---|---|
| 1_sim_method_schematic. ipynb | Simulations: *Figure 1B–E* |
| 2_viz_NeuroTycho-SU. ipynb | Macaque timescales: *Figure 2E–G*, *Figure 2—figure supplement 4* |
| 3_viz_human_structural. ipynb | Human timescales vs. T1w/T2w and gene expression: *Figure 2A–D*, *Figure 2—figure supplements 1* and *3*, *Figure 3*, *Figure 3—figure supplements 1* and *2*, *Supplementary file 1–*, *Supplementary file 2*, *Supplementary file 3*. |
| 4b_viz_human_wm.ipynb | Human working memory: *Figure 4A–D*, *Figure 4—figure supplement 1* |
| 4a_viz_human_aging. ipynb | Human aging: *Figure 4E and F*, *Figure 4—figure supplement 2* |
| supp_spatialautocorr. ipynb | Spatial autocorrelation-preserving nulls: |
| supp_spatialautocorr. ipynb | Spatial autocorrelation-preserving nulls: *Figure 2—figure supplement 2* |

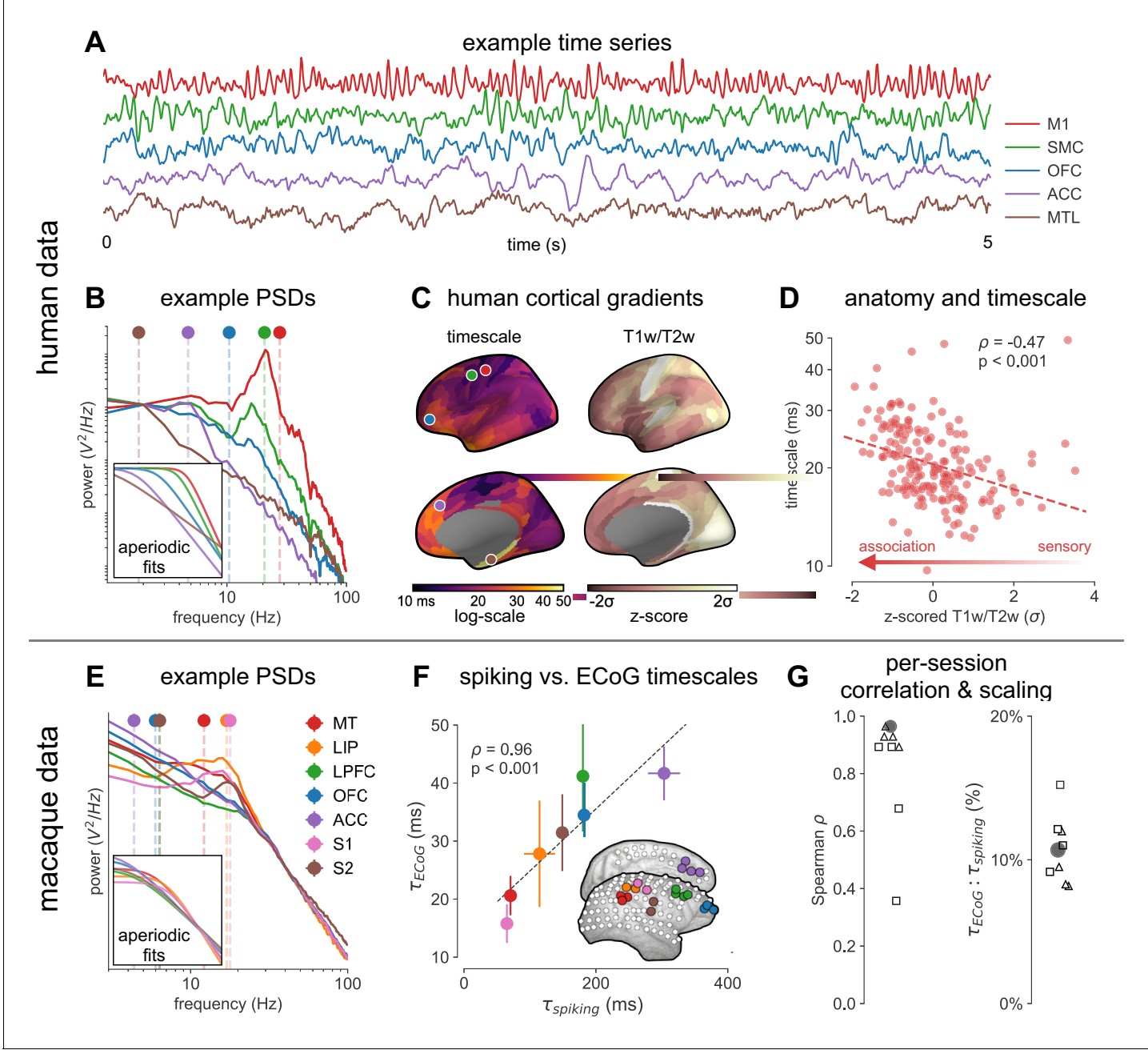

**Figure 2.** Timescale increases along the anatomical hierarchy in humans and macaques. (A) Example time series from five electrodes along the human cortical hierarchy (M1: primary motor cortex; SMC: supplementary motor cortex; OFC: orbitofrontal cortex; ACC: anterior cingulate cortex; MTL: medial temporal lobe), and (B) their corresponding power spectral densities (PSDs) computed over 1 min. Circle and dashed line indicate the knee frequency for each PSD, derived from the aperiodic component fits (inset). Data: MNI-iEEG database, N = 106 participants. (C) Human cortical timescale gradient (left) falls predominantly along the rostrocaudal axis, similar to T1w/T2w ratio (right; z-scored, in units of standard deviation). Colored dots show electrode locations of example data. (D) Neuronal timescales are negatively correlated with cortical T1w/T2w, thus increasing along the anatomical hierarchy from sensory to association regions (Spearman correlation; p-value corrected for spatial autocorrelation, *Figure 2—figure supplement 2A–C*). (E) Example PSDs from macaque ECoG recordings, similar to (B) (LIP: lateral intraparietal cortex; LPFC: lateral prefrontal cortex; S1 and S2: primary and secondary somatosensory cortex). PSDs are averaged over electrodes within each region (inset of [F]). Data: Neurotycho, N = 8 sessions from two animals. (F) Macaque ECoG timescales track published single-unit spiking timescales (*Murray et al., 2014*) in corresponding regions (error bars represent mean ± s.e.m.). Inset: ECoG electrode map of one animal and selected electrodes for comparison. (G) ECoG-derived timescales are consistently correlated with (left), and ~10 times faster than (right), single-unit timescales across individual sessions. Hollow markers: individual sessions; shapes: animals; solid circles: grand average from (F).

The online version of this article includes the following figure supplement(s) for figure 2:

*Figure 2 continued on next page*

**Figure supplement 1.** MNI-iEEG dataset electrode coverage.

**Figure supplement 2.** Comparison of spatial autocorrelation-preserving null map generation methods.

**Figure supplement 3.** Cortical thickness.

**Figure supplement 4.** Macaque ECoG and single-unit coverage.

either surface strip/grid or stereoEEG electrodes, and cleaned of visible artifacts. *Figure 2A* shows example data traces along the cortical hierarchy with increasing timescales estimated from their PSDs (*Figure 2B*; circles denote fitted knee frequency). Timescales from individual channels were extracted and projected from MNI coordinates onto the left hemisphere of HCP-MMP1.0 surface parcellation (*Glasser et al., 2016*) for each patient using a Gaussian-weighted mask centered on each electrode. While coverage is sparse and idiosyncratic in individual patients, it does not vary as a function of age, and when pooling across the entire population, 178 of 180 parcels have at least one patient with an electrode within 4 mm (*Figure 2—figure supplement 1A–F*).

Across the human cortex, timescales of fast electrophysiological dynamics (~10–50 ms) predominantly follow a rostrocaudal gradient (*Figure 2C*, circles denote location of example data from 2A). Consistent with numerous accounts of a principal cortical axis spanning from primary sensory to association regions (*Hilgetag and Goulas, 2020*; *Margulies et al., 2016*; *Wang, 2020*), timescales are shorter in sensorimotor and early visual areas, and longer in association regions, especially cingulate, ventral/medial frontal, and medial temporal lobe (MTL) regions (*Figure 2—figure supplement 1G* shows further pooling into 21 labeled macro-regions). We then compare the timescale gradient to the average T1w/T2w map from the Human Connectome Project, which captures gray matter myelination and indexes the proportion of feedforward vs. feedback connections between cortical regions, thus acting as a noninvasive proxy of connectivity-based anatomical hierarchy (*Burt et al., 2018*; *Glasser and Van Essen, 2011*). We find that neuronal timescales are negatively correlated with T1w/T2w across the entire cortex (*Figure 2D*, $\rho = -0.47$, p<0.001; corrected for spatial autocorrelation [SA], see Materials and methods and *Figure 2—figure supplement 2A–C* for a comparison of correction methods), such that timescales are shorter in more heavily myelinated (i.e., lower-level, sensory) regions. Timescales are also positively correlated with cortical thickness (*Figure 2—figure supplement 3*, $\rho = 0.37$, p=0.035)—another index of cortical hierarchy that is itself anti-correlated with T1w/T2w. Thus, we observe that neuronal timescales lengthen along the human cortical hierarchy, from sensorimotor to association regions.

While surface ECoG recordings offer much broader spatial coverage than extracellular single-unit recordings, they are fundamentally different signals: ECoG and field potentials largely reflect integrated synaptic and other transmembrane currents across many neuronal and glial cells, rather than putative action potentials from single neurons (*Buzsáki et al., 2012*; *Figure 1A*, yellow box). Considering this, we ask whether timescales measured from ECoG in this study ($\tau_{ECoG}$) are related to single-unit spiking timescales along the cortical hierarchy ($\tau_{spiking}$). To test this, we extract neuronal timescales from task-free ECoG recordings in macaques (*Nagasaka et al., 2011*) and compare them to a separate dataset of single-unit spiking timescales from a different group of macaques (*Murray et al., 2014*) (see *Figure 2—figure supplement 4* for electrode locations). Consistent with $\tau_{spiking}$ estimates (*Murray et al., 2014*; *Wasmuht et al., 2018*), $\tau_{ECoG}$ also increase along the macaque cortical hierarchy. While there is a strong correspondence between spiking and ECoG timescales (*Figure 2F*; $\rho = 0.96$, p<0.001)—measured from independent datasets—across the macaque cortex, $\tau_{ECoG}$ are ~10 times faster than $\tau_{spiking}$ and are conserved across individual sessions (*Figure 2G*). This suggests that neuronal spiking and transmembrane currents have distinct but related timescales of fluctuations, and that both are hierarchically organized along the primate cortex.

## Synaptic and ion channel genes shape timescales of neuronal dynamics

Next, we identify potential cellular and synaptic mechanisms underlying timescale variations across the human cortex. Theoretical accounts posit that NMDA-mediated recurrent excitation coupled with fast inhibition (*Chaudhuri et al., 2015*; *Wang, 2008*; *Wang, 1999*), as well as cell-intrinsic properties (*Duarte and Morrison, 2019*; *Gjorgjieva et al., 2016*; *Koch et al., 1996*), are crucial for

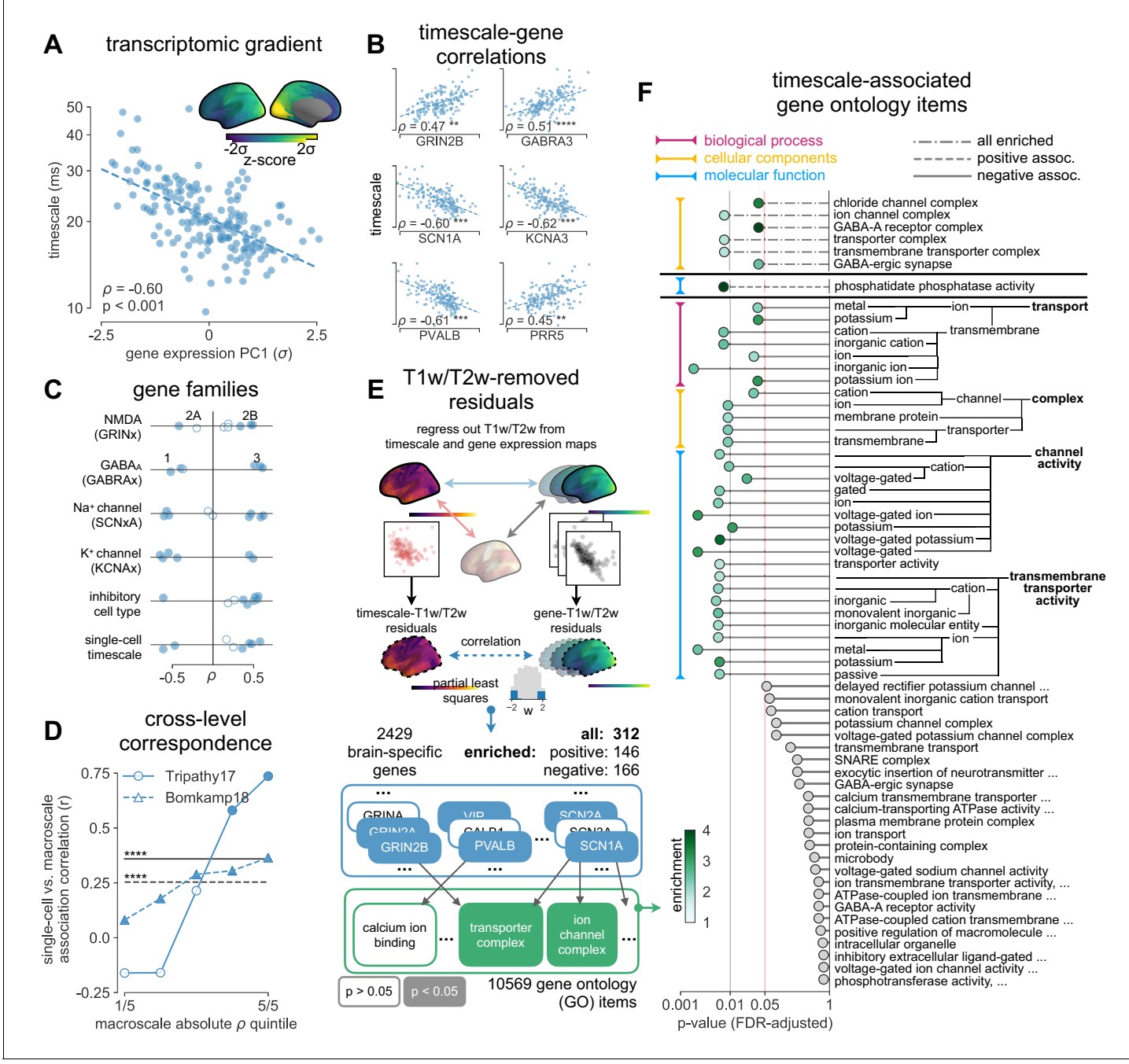

**Figure 3.** Timescale gradient is linked to expression of genes related to synaptic receptors and transmembrane ion channels across the human cortex. (A) Timescale gradient follows the dominant axis of gene expression variation across the cortex (z-scored PC1 of 2429 brain-specific genes, arbitrary direction). (B) Timescale gradient is significantly correlated with expression of genes known to alter synaptic and neuronal membrane time constants, as well as inhibitory cell-type markers, but (C) members within a gene family (e.g., NMDA receptor subunits) can be both positively and negatively associated with timescales, consistent with predictions from in vitro studies. (D) Macroscale timescale-transcriptomic correlation captures association between RNA-sequenced expression of the same genes and single-cell timescale properties fit to patch clamp data from two studies, and the correspondence improves for genes (separated by quintiles) that are more strongly correlated with timescale (solid: N = 170 [*Tripathy et al., 2017*], dashed: N = 4168 genes [*Bomkamp et al., 2019*]; horizontal lines: correlation across all genes from the two studies, $\rho$ = 0.36 and 0.25, p<0.001 for both). (E) T1w/T2w gradient is regressed out from timescale and gene expression gradients, and a partial least squares (PLS) model is fit to the residual maps. Genes with significant PLS weights (filled blue boxes) compared to spatial autocorrelation (SA)-preserved null distributions are submitted for gene ontology enrichment analysis (GOEA), returning a set of significant GO terms that represent functional gene clusters (filled green boxes). (F) Enriched genes are primarily linked to potassium and chloride transmembrane transporters, and GABA-ergic synapses; genes specifically with strong negative associations further over-represent transmembrane ion exchange mechanisms, especially voltage-gated potassium and cation transporters.
*Figure 3 continued on next page*

*Figure 3 continued*

Branches indicate GO items that share higher-level (parent) items, e.g., *voltage-gated cation channel activity* is a child of *cation channel activity* in the molecular functions (MF) ontology, and both are significantly associated with timescale. Color of lines indicate curated ontology (BP—biological process, CC—cellular components, or MF). Dotted, dashed, and solid lines correspond to analysis performed using all genes or only those with positive or negative PLS weights. Spatial correlation p-values in (A–C) are corrected for SA (see Materials and methods; asterisks in (B,D) indicate p<0.05, 0.01, 0.005, and 0.001 respectively; filled markers in (C,D) indicate p<0.05.

The online version of this article includes the following figure supplement(s) for figure 3:

**Figure supplement 1.** Transcriptomic principal component analysis results.
**Figure supplement 2.** Individual timescale-gene correlations magnitudes.

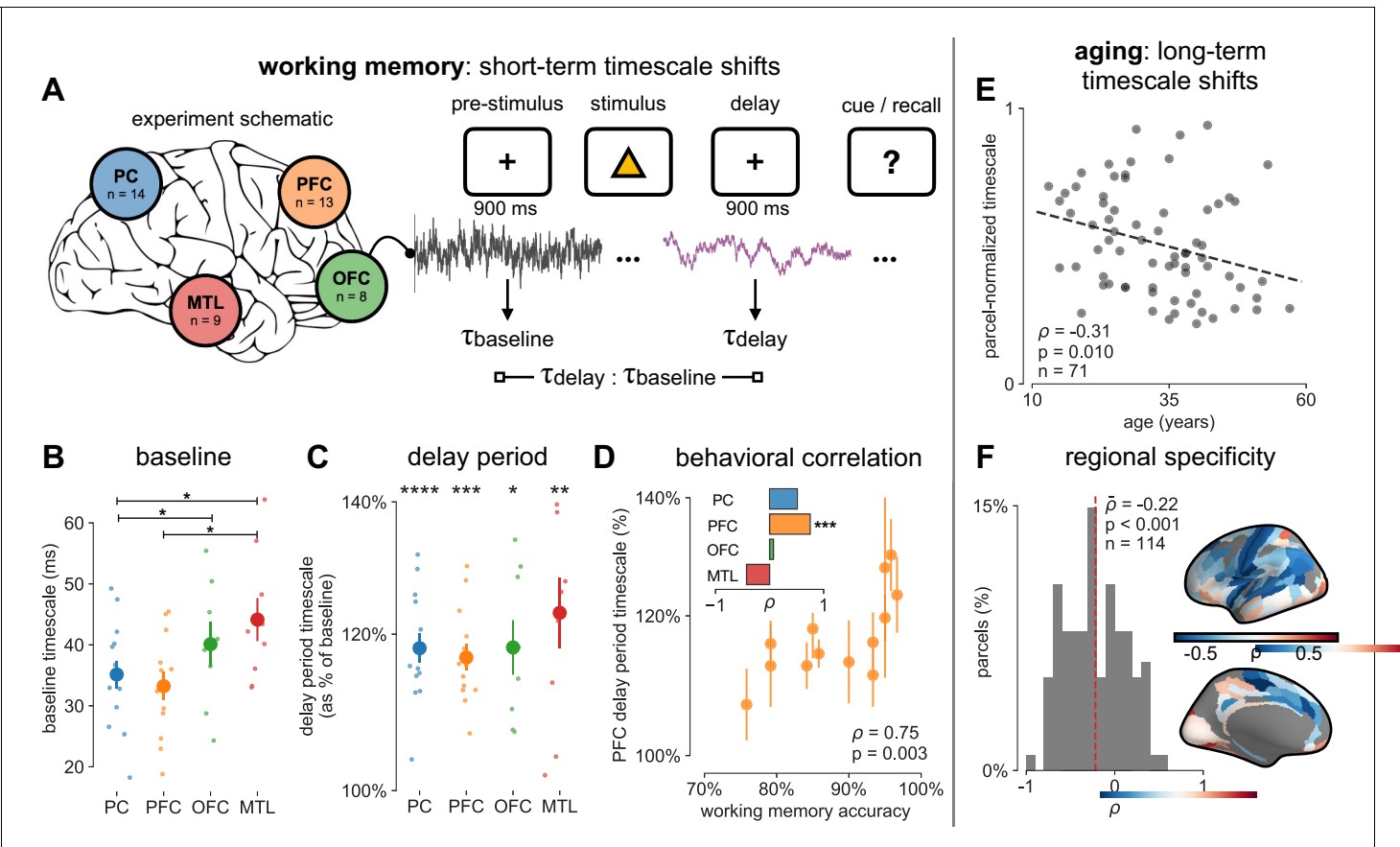

**Figure 4.** Timescales expand during working memory maintenance while tracking performance, and task-free average timescales compress in older adults. (A) Fourteen participants with overlapping intracranial coverage performed a visuospatial working memory task, with 900 ms of baseline (pre-stimulus) and delay period data analyzed (PC: parietal, PFC: prefrontal, OFC: orbitofrontal, MTL: medial temporal lobe; n denotes number of subjects with electrodes in that region). (B) Baseline timescales follow hierarchical organization within association regions (*: p<0.05, Mann–Whitney U-test; small dots represent individual participants, large dots and error bar for mean ± s.e.m. across participants). (C) All regions show significant timescale increase during delay period compared to baseline (asterisks represent p<0.05, 0.01, 0.005, 0.001, Wilcoxon signed-rank test). (D) PFC timescale expansion during delay periods predicts average working memory accuracy across participants (dot represents individual participants, mean ± s.e.m. across PFC electrodes within participant); inset: correlation between working memory accuracy and timescale change for all regions. (E) In the MNI-iEEG dataset, participant-average cortical timescales decrease (become faster) with age (n = 71 participants with at least 10 valid parcels, see *Figure 4—figure supplement 2B*). (F) Most cortical parcels show a negative relationship between timescales and age, with the exception being parts of the visual cortex and the temporal poles (one-sample t-test, $t = −7.04$, p<0.001; n = 114 parcels where at least six participants have data, see *Figure 4—figure supplement 2C*).

The online version of this article includes the following figure supplement(s) for figure 4:

**Figure supplement 1.** Spectral correlates of working memory performance.
**Figure supplement 2.** Parameter sensitivity for timescale-aging analysis.

shaping neuronal timescales. While in vitro and in vivo studies in model organisms (*van Vugt et al., 2020*; *Wang et al., 2013*) can test these hypotheses at the single-neuron level, causal manipulations and large-scale recordings of neuronal networks embedded in the human brain are severely limited. Here, we apply an approach analogous to multimodal single-cell profiling (*Bomkamp et al., 2019*) and examine the transcriptomic basis of neuronal dynamics at the macroscale.

Leveraging whole-cortex interpolation of the Allen Human Brain Atlas bulk mRNA expression (*Gryglewski et al., 2018*; *Hawrylycz et al., 2012*), we project voxel-wise expression maps onto the HCP-MMP1.0 surface parcellation, and find that the neuronal timescale gradient overlaps with the dominant axis of gene expression (i.e., first principal component of 2429 brain-related genes) across the human cortex (*Figure 3A*, $\rho = -0.60$, p<0.001; see *Figure 3—figure supplement 1* for similar results with all 18,114 genes). Consistent with theoretical predictions (*Figure 3B*), timescales significantly correlate with the expression of genes encoding for NMDA (GRIN2B) and GABA-A (GABRA3) receptor subunits, voltage-gated sodium (SCN1A) and potassium (KCNA3) ion channel subunits, as well as inhibitory cell-type markers (parvalbumin, PVALB), and genes previously identified to be associated with single-neuron membrane time constants (PRR5) (*Bomkamp et al., 2019*) (all Spearman correlations corrected for SA in gradients).

More specifically, in vitro electrophysiological studies have shown that, for example, increased expression of receptor subunit 2B extends the NMDA current time course (*Flint et al., 1997*), while 2A expression shortens it (*Monyer et al., 1994*). Similarly, the GABA-A receptor time constant lengthens with increasing a3:a1 subunit ratio (*Eyre et al., 2012*). We show that these relationships are recapitulated at the macroscale, where neuronal timescales positively correlate with GRIN2B and GABRA3 expression and negatively correlate with GRIN2A and GABRA1 (*Figure 3C*). These results demonstrate that timescales of neural dynamics depend on specific receptor subunit combinations with different (de)activation timescales (*Duarte et al., 2017*; *Gjorgjieva et al., 2016*), in addition to broad excitation–inhibition interactions (*Gao et al., 2017*; *Wang, 2020*; *Wang, 2002*). Notably, almost all genes related to voltage-gated sodium and potassium ion channel alpha-subunits—the main functional subunits—are correlated with timescale, while all inhibitory cell-type markers except parvalbumin have strong positive associations with timescale (*Figure 3C* and *Figure 3—figure supplement 2*).

We further test whether single-cell timescale-transcriptomic associations are captured at the macroscale as follows: for a given gene, we can measure how strongly its expression correlates with membrane time constant parameters at the single-cell level using patch-clamp and RNA sequencing (scRNASeq) data (*Bomkamp et al., 2019*; *Tripathy et al., 2017*). Analogously, we can measure its macroscopic transcriptomic-timescale correlation using the cortical gradients above. If the association between the expression of this gene and neuronal timescale is preserved at both levels, then the correlation across cells at the microscale should be similar to the correlation across cortical regions at the macroscale. Comparing across these two scales for all previously identified timescale-related genes in two studies (N = 170 [*Tripathy et al., 2017*] and 4168 [*Bomkamp et al., 2019*] genes), we find a significant correlation between the strength of association at the single-cell and macroscale levels (*Figure 3D*, horizontal black lines; $\rho = 0.36$ and 0.25 for the two datasets, p<0.001 for both). Furthermore, genes with stronger associations to timescale tend to conserve this relationship across single-cell and macroscale levels (*Figure 3D*, separated by macroscale correlation magnitude). Thus, the association between cellular variations in gene expression and cell-intrinsic temporal dynamics is captured at the macroscale, even though scRNAseq and microarray data represent entirely different measurements of gene expression.

While we have shown associations between cortical timescales and genes suspected to influence neuronal dynamics, these data present an opportunity to discover additional novel genes that are functionally related to timescales through a data-driven approach. However, since transcriptomic variation and anatomical hierarchy overlap along a shared macroscopic gradient (*Burt et al., 2018*; *Huntenburg et al., 2018*; *Margulies et al., 2016*), we cannot specify the role certain genes play based on their level of association with timescale alone: gene expression differences across the cortex first result in cell-type and connectivity differences, sculpting the hierarchical organization of cortical anatomy. Consequently, anatomy and cell-intrinsic properties jointly shape neuronal dynamics through connectivity differences (*Chaudhuri et al., 2015*; *Demirtaş et al., 2019*) and expression of ion transport and receptor proteins with variable activation timescales, respectively. Therefore, we ask whether variation in gene expression still accounts for variation in timescale beyond the principal

structural gradient, and if associated genes have known functional roles in biological processes (BP) (schematic in *Figure 3E*). To do this, we first remove the contribution of anatomical hierarchy by linearly regressing out the T1w/T2w gradient from both timescale and individual gene expression gradients. We then fit partial least squares (PLS) models to simultaneously estimate regression weights for all genes (*Whitaker et al., 2016*), submitting those with significant associations for gene ontology enrichment analysis (GOEA) (*Klopfenstein et al., 2018*).

We find that genes highly associated with neuronal timescales are preferentially related to transmembrane ion transporter complexes, as well as GABAergic synapses and chloride channels (see *Figure 3F* and *Supplementary file 1* for GOEA results with brain genes only, and *Supplementary file 2* for all genes). When restricted to positively associated genes only (expression increases with timescales), one functional group related to phosphatidate phosphatase activity is uncovered, including the gene PLPPR1, which has been linked to neuronal plasticity (*Savaskan et al., 2004*)—a much slower timescale physiological process. Conversely, genes that are negatively associated with timescale are related to numerous groups involved in the construction and functioning of transmembrane transporters and voltage-gated ion channels, especially potassium and other inorganic cation transporters. To further ensure that these genes specifically relate to neuronal timescale, we perform the same enrichment analysis with T1w/T2w vs. gene maps as a control. The control analysis yields no significant GO terms when restricted to brain-specific genes (in contrast to *Figure 3F*), while repeating the analysis with all genes does yield significant GO terms related to ion channels and synapses, but are much less specific to those (see *Supplementary file 3*), including a variety of other gene clusters associated with general metabolic processes, signaling pathways, and cellular components (CC). This further strengthens the point that removing the contribution of T1w/T2w aids in identifying genes that are more specifically associated with neurodynamics, suggesting that inhibition (*Teleńczuk et al., 2017*)—mediated by GABA and chloride channels— and voltage-gated potassium channels have prominent roles in shaping neuronal timescale dynamics at the macroscale level, beyond what is expected based on the anatomical hierarchy alone.

## Timescales lengthen in working memory and shorten in aging

Finally, having shown that neuronal timescales are associated with stable anatomical and gene expression gradients across the human cortex, we turn to the final question of the study: are cortical timescales relatively static, or are they functionally dynamic and relevant for human cognition? While previous studies have shown hierarchical segregation of task-relevant information corresponding to intrinsic timescales of different cortical regions (*Baldassano et al., 2017*; *Chien and Honey, 2020*; *Honey et al., 2012*; *Runyan et al., 2017*; *Sarafyazd and Jazayeri, 2019*; *Wasmuht et al., 2018*), as well as optimal adaptation of behavioral timescales to match the environment (*Ganupuru et al., 2019*; *Ossmy et al., 2013*), evidence for functionally relevant changes in regional neuronal timescales is lacking. Here, we examine whether timescales undergo short- and long-term shifts during working memory maintenance and aging, respectively.

We first analyze human ECoG recordings from parietal, frontal (PFC and orbitofrontal cortex [OFC]), and medial temporal lobe (MTL) regions of patients (N = 14) performing a visuospatial working memory task that requires a delayed cued response (*Figure 4A*; *Johnson et al., 2018a*). Neuronal timescales were extracted for pre-stimulus baseline and memory maintenance delay periods (900 ms, both stimulus-free). Replicating our previous result in *Figure 2—figure supplement 1G*, we observe that baseline neuronal timescales follow a hierarchical progression across association regions, where parietal cortex (PC), PFC, OFC, and MTL have gradually longer timescales (pairwise Mann–Whitney U-test, *Figure 4B*). If neuronal timescales track the temporal persistence of information in a functional manner, then they should expand during delay periods. Consistent with our prediction, timescales in all regions are ~20% longer during delay periods (*Figure 4C*; Wilcoxon rank-sum test). Moreover, only timescale changes in the PFC are significantly correlated with behavior across participants, where longer delay-period timescales relative to baseline are associated with better working memory performance (*Figure 4D*, $\rho = 0.75$, p=0.003). No other spectral features in the recorded brain regions show consistent changes from baseline to delay periods while also significantly correlating with individual performance, including the 1/f-like spectral exponent, narrowband theta (3–8 Hz), and high-frequency (high gamma; 70–100 Hz) activity power (*Figure 4—figure supplement 1*).

While timescales are consistent with the anatomical and gene expression hierarchy at a snapshot, brain structure itself is not static over time, undergoing many slower, neuroplastic changes during early development and throughout aging in older populations. In particular, aging is associated with a broad range of functional and structural changes, such as working memory impairments (*Voytek et al., 2015c*; *Wang et al., 2011*), as well as changes in neuronal dynamics (*Voytek et al., 2015c*; *Voytek and Knight, 2015b*; *Wang et al., 2011*) and cortical structure (*de Villers-Sidani et al., 2010*), such as the loss of slow-deactivating NMDA receptor subunits (*Pegasiou et al., 2020*). Since neuronal timescales may support working memory maintenance, we predict that timescales would shorten across the lifespan, in agreement with the observed cognitive and structural deteriorations. To this end, we leverage the wide age range in the MNI-iEEG dataset (13–62 years old) and probe average cortical timescales for each participant as a function of age. Since ECoG coverage is sparse and nonuniform across participants, simply averaging across parcels within individual participants confounds the effect of aging with the spatial effect of cortical hierarchy. Instead, we first normalize each parcel by its max value across all participants before averaging within participants, excluding those with fewer than 10 valid parcels (N = 71 of 106 subjects remaining; results hold for a large range of threshold values, *Figure 4—figure supplement 2B*). We observe that older adults have faster neuronal timescales ($\rho = -0.31$, p=0.010; *Figure 4E*), and that timescales shorten with age in most areas across the cortex (*Figure 4F*, $t = -7.04$, p<0.001; 114 out of 189 parcels where at least six participants have data, see *Figure 4—figure supplement 2C*). This timescale compression is especially prominent in sensorimotor, temporal, and medial frontal regions. These results support our hypothesis that neuronal timescales, estimated from transmembrane current fluctuations, can rapidly shift in a functionally relevant manner, as well as slowly—over decades—in healthy aging.

## Discussion

Theoretical accounts and converging empirical evidence predict a graded variation of neuronal timescales across the human cortex (*Chaudhuri et al., 2015*; *Huntenburg et al., 2018*; *Wang, 2020*), which reflects functional specialization and implements hierarchical temporal processing crucial for complex cognition (*Kiebel et al., 2008*). This timescale gradient is thought to emerge as a consequence of cortical variations in cytoarchitecture, as well as both macroscale and microcircuit connectivity, thus serving as a bridge from brain structure to cognitive function (*Kanai and Rees, 2011*). In this work, we infer the timescale of non-rhythmic transmembrane current fluctuations from invasive human intracranial recordings and test those predictions explicitly. We discuss the implications and limitations of our findings below.

### Multiple quantities for neuronal timescale and anatomical hierarchy

We first find that neuronal timescales vary continuously across the human cortex and coincide with the anatomical hierarchy, with timescales increasing from primary sensory and motor to association regions. While we use the continuous T1w/T2w gradient as a surrogate measure for anatomical hierarchy, there are multiple related but distinct perspectives on what 'cortical hierarchy' means, including, for example, laminar connectivity patterns from tract tracing data (*Felleman and Van Essen, 1991*; *Vezoli et al., 2020*), continuous (and latent-space) gradients of gene expression and microarchitectural features (*Huntenburg et al., 2018*), and network connectivity scales (see review of *Hilgetag and Goulas, 2020*)—with most of these following a graded sensorimotor-to-association area progression. Similarly, it is important to note that there exist many different quantities that can be considered as characteristic neuronal timescales across several spatial scales, including membrane potential and synaptic current timescales (*Duarte et al., 2017*), single-unit spike-train timescales (*Murray et al., 2014*), population code timescales (*Runyan et al., 2017*), and even large-scale circuit timescales measured from the fMRI BOLD signal (*Watanabe et al., 2019*). We show here that timescales inferred from ECoG are consistently approximately 10 times faster than single-unit spiking timescales in macaques, corroborating the fact that field potential signals mainly reflect fast transmembrane and synaptic currents (*Buzsáki et al., 2012*), whose timescales are related to, but distinct from, single-unit timescales measured in previous studies (*Dotson et al., 2018*; *Ogawa and Komatsu, 2010*; *Wasmuht et al., 2018*).

Because field potential fluctuations are driven by currents from both locally generated and distal inputs, our results raise questions on how and when these timescales interact to shape downstream

spiking dynamics. Furthermore, while we specifically investigate here the aperiodic timescale, which corresponds to the exponential decay timescale measured in previous studies, recent work has shown a similar gradient of oscillatory timescales (i.e., frequency) along the anterior–posterior axis of the human cortex (*Mahjoory et al., 2020*). Based on the similarity of these gradients and known mechanisms of asynchronous and oscillatory population dynamics (e.g., balance of excitation and inhibition in generating gamma oscillations and the asynchronous irregular state in cortical circuits [*Brunel, 2000*; *Brunel and Wang, 2003*]), we speculate that timescales of oscillatory and aperiodic neural dynamics may share (at least partially) circuit mechanisms at different spatial scales, analogous to the relationship between characteristic frequency and decay constant in a damped harmonic oscillator model.

## Collinearity and surrogate nature of postmortem gene expression gradients

Using postmortem gene expression data as a surrogate for protein density, transcriptomic analysis uncovers the potential roles that transmembrane ion transporters and synaptic receptors play in establishing the cortical gradient of neuronal timescales. The expression of voltage-gated potassium channel, chloride channel, and GABAergic receptor genes, in particular, are strongly associated with the spatial variation of neuronal timescale. Remarkably, we find that electrophysiology-transcriptomic relationships discovered at the single-cell level, through patch-clamp recordings and single-cell RNA sequencing, are recapitulated at the macroscale between bulk gene expression and timescales inferred from ECoG. That being said, it is impossible to make definitive causal claims with the data presented in this study, especially considering the fact that several microanatomical features, such as gray matter myelination and cortical thickness, follow similar gradients across the cortex (*Burt et al., 2018*). To discover genes specifically associated with timescale while accounting for the contribution of the overlapping anatomical hierarchy, we linearly regress out the T1w/T2w gradient from both timescale and gene expression gradients. Although this procedure does not account for any nonlinear contributions from anatomy, gene enrichment control analysis using T1w/T2w instead of timescales further demonstrates that the discovered genes—transmembrane ion transporters and inhibitory synaptic receptors—are more specifically associated with the timescale gradient, over and above the level predicted by anatomical hierarchy alone. From these results, we infer that potassium and chloride ion channels, as well as GABAergic receptors, may play a mechanistic role in altering the timescale of transmembrane currents at the macroscopic level.

However, this interpretation rests on the key assumption that mRNA expression level is a faithful representation of the amount of functional proteins in a given brain region. In general, gene expression levels are highly correlated with the percentage of cells expressing that gene within brain regions (*Lein et al., 2007*). Therefore, on a population level, the regional density of a particular ion channel or receptor protein is high if bulk mRNA expression is high. Furthermore, recent works have shown that neurotransmitter receptor density measured via autoradiography in postmortem brains follows similar cortical gradients (*Goulas et al., 2020*), and that gene expression levels of neurotransmitter receptors (e.g., 5HT) are strongly correlated with ligand binding potential measured via PET (*Gryglewski et al., 2018*). Thus, as a first order approximation, receptor gene expression is an adequate surrogate for receptor protein density in the brain at the macroscale, though the relationship between mRNA expression and their transport and translation into channel proteins, the process of incorporating those proteins into membranes and synapses, and how these gene expression maps can be related to other overlapping macroscopic gradients are complex issues (see e.g., *Fornito et al., 2019*; *Liu et al., 2016*). Thus, our analyses represent an initial data-mining process at the macroscopic level, which should motivate further studies in investigating the precise roles voltage-gated ion channels and synaptic inhibition play in shaping functional neuronal timescales through causal manipulations, complementary to existing lines of research focusing on NMDA activation and recurrent circuit motifs.

## Structural constraints vs. behaviorally required flexibility in timescale

Finally, we show that neuronal timescales are not static, but can change both in the short and long terms. Transmembrane current timescales across multiple association regions, including parietal, frontal, and medial temporal cortices, increase during the delay period of a working memory

task, consistent with the emergence of persistent spiking during working memory delay. Working memory performance across individuals, however, is predicted by the extent of timescale increase in the PFC only. This further suggests that behaviorally relevant neural activity may be localized despite widespread task-related modulation (*Pinto et al., 2019*), even at the level of neuronal membrane fluctuations. In the long term, we find that neuronal timescale shortens with age in most cortical regions, linking age-related synaptic, cellular, and connectivity changes—particularly those that influence neuronal integration timescale—to the compensatory posterior-to-anterior shift of functional specialization in healthy aging (*Davis et al., 2008*).

These results raise further questions regarding contrasting, and potentially complementary, aspects of neuronal timescale: on the one hand, task-free timescales across the cortex are shaped by relatively static macro- and microarchitectural properties (*Figures 2* and *3*); on the other hand, timescales are dynamic and shift with behavioral demand (*Figure 4*). While long-term structural changes in the brain can explain shifts in neuronal timescales throughout the aging process, properties such as ion channel protein density probably do not change within seconds during a working memory task. We speculate that structural properties may constrain dynamical properties (such as timescale) to a possible range within a particular brain region and at different spatial scales, while task requirements, input statistics, short-term synaptic plasticity, and neuromodulation can then shift timescale within this range. We posit, then, that only shifts in dynamics within the area of relevance (i.e., PFC for working memory) are indicative of task performance, consistent with recent ideas of computation-through-dynamics (*Vyas et al., 2020*). Nevertheless, which neuromodulatory and circuit mechanisms are involved in shifting local timescales, and how timescales at different spatial scales (e.g., synaptic, neuronal, population) interact to influence each other remain open questions for future investigation (*Breakspear, 2017*; *Duarte et al., 2017*; *Freeman, 2000*; *Freeman and Erwin, 2008*; *Gjorgjieva et al., 2016*; *Shine et al., 2019*).

## Conclusion

In summary, we identify consistent and converging patterns between transcriptomics, anatomy, dynamics, and function across multiple datasets of different modalities from different individuals and multiple species. As a result, evidence for these relationships can be supplemented by more targeted approaches such as imaging of receptor metabolism. Furthermore, the introduction and validation of an open-source toolbox (*Donoghue et al., 2020*) for inferring timescales from macroscale electrophysiological recordings potentially allows for the noninvasive estimation of neuronal timescales, using widely accessible tools such as EEG and MEG. These results open up many avenues of research for discovering potential relationships between microscale gene expression and anatomy with the dynamics of neuronal population activity at the macroscale in humans.

## Materials and methods

### Inferring timescale from autocorrelation and PSD

Consistent with previous studies, we define 'neuronal timescale' as the exponential decay time constant ($\tau$) of the empirical ACF, or lagged correlation (*Honey et al., 2012*; *Murray et al., 2014*). $\tau$ can be naively estimated to be the time it takes for the ACF to decrease by a factor of $e$ when there are no additional long-term, scale-free, or oscillatory processes, or by fitting a function of the form $f(t) = e^{-\frac{t}{\tau}}$ and extracting the parameter $\tau$. Equivalently, the PSD is the Fourier Transform of the ACF via Wiener–Khinchin theorem (*Khintchine, 1934*) and follows a Lorentzian function of the form $L(f) = \frac{A}{k+f^{\chi}}$ for approximately exponential-decay processes, with $\chi = 2$ exactly when the ACF is solely composed of an exponential decay term, though it is often variable and in the range between 2 and 6 for neural time series (*Donoghue et al., 2020*; *Miller et al., 2009*; *Podvalny et al., 2015*; *Voytek et al., 2015c*). Timescale can be computed from the parameter $k$ as $\tau = \frac{1}{2\pi f_k}$, where $f_k \approx k^{1/\chi}$ is approximated to be the 'knee frequency', at which a bend or knee in the power spectrum occurs, and equality holds when $\chi = 2$.

## Computing PSD

PSDs are estimated using a modified Welch's method, where short-time windowed Fourier transforms (STFT) are computed from the time series, but the median is taken across time instead of the mean (in conventional Welch's method) to minimize the effect of high-amplitude transients and artifacts (*Izhikevich et al., 2018*). Custom functions for this can be found in NeuroDSP (*Cole et al., 2019*), a published and open-source digital signal processing toolbox for neural time series (neurodsp.spectral.compute_spectrum). For simulated data, Neurotycho macaque ECoG, and MNI-iEEG datasets, we use 1 s long Hamming windows with 0.5 s overlap. To estimate single-trial PSDs for the working memory ECoG dataset (CRCNS Johnson-ECoG *Johnson et al., 2018a*; *Johnson et al., 2018b*), we simply apply Hamming window to 900 ms long epoched time series and compute the squared magnitude of the windowed Fourier transform.

## Spectral parametrization

We apply spectral parameterization (*Donoghue et al., 2020*) to extract timescales from PSDs. Briefly, we decompose log-power spectra into a summation of narrowband periodic components—modeled as Gaussians—and an aperiodic component—modeled as a generalized Lorentzian function centered at 0 Hz ($L(f)$ above). For inferring decay timescale, this formalism can be practically advantageous when a strong oscillatory or variable power-law ($\chi$) component is present, as is often the case for neural signals. While oscillatory and power-law components can corrupt naive measurements of $\tau$ as time for the ACF to reach $1/e$, they can be easily accounted for and ignored in the frequency domain as narrowband peaks and 1/f-exponent fit. We discard the periodic components and infer timescale from the aperiodic component of the PSD. For a complete mathematical description of the model, see *Donoghue et al., 2020*.

## Simulation and validation

We simulate the aperiodic background component of neural field potential recordings as autocorrelated stochastic processes by convolving Poisson population spikes with exponentially decaying synaptic kernels with predefined decay time constants (neurodsp.sim.sim_synaptic_current). PSDs of the simulated data are computed and parameterized as described above, and we compare the fitted timescales with their ground-truth values.

## Macaque ECoG and single-unit timescales data

Macaque single-unit timescales are taken directly from values reported in Figure 1c of *Murray et al., 2014*. Whole-brain surface ECoG data (1000 Hz sampling rate) is taken from the Neurotycho repository (*Nagasaka et al., 2011*; *Yanagawa et al., 2013*), with eight sessions of 128-channel recordings from two animals (George and Chibi, four sessions each). Results reported in *Figure 2E–G* are from ~10 min eyes-open resting periods to match the pre-stimulus baseline condition of single-unit experiments. Timescales for individual ECoG channels are extracted and averaged over regions corresponding to single-unit recording areas from *Murray et al., 2014*; *Figure 2F* inset and *Figure 2—figure supplement 3*, which are selected visually based on the overlapping cortical map and landmark sulci/gyri. Each region included between two and four electrodes (see *Figure 2—figure supplement 3B* for selected ECoG channel indices for each region).

## Statistical analysis for macaque ECoG and spiking timescale

For each individual recording session, as well as the grand average, Spearman rank correlation was computed between spiking and ECoG timescales. Linear regression models were fit using the python package scipy (*Virtanen et al., 2020*) (scipy.stats.linregress) and the linear slope was used to compute the scaling coefficient between spiking and ECoG timescales.

## Variations in neuronal timescale, T1/T2 ratio, and mRNA expression across human cortex

The following sections describe procedures for generating the average cortical gradient maps for neuronal timescale, MR-derived T1w/T2w ratio, and gene expression from the respective raw datasets. All maps were projected onto the 180 left hemisphere parcels of Human Connectome Project's Multimodal Parcellation (*Glasser et al., 2016*) (HCP-MMP1.0) for comparison, described in the

individual sections. Projection of T1w/T2w and gene expression maps from MNI volumetric coordinates to HCP-MMP1.0 can be found: https://github.com/rudyvdbrink/Surface_projection (*van den Brink, 2020*).

All spatial correlations are computed as Spearman rank correlations between maps. Procedure for computing statistical significance while accounting for SA is described in detail below under the sections 'Spatial statistics' and 'SA modeling'.

## Neuronal timescale map

The MNI Open iEEG dataset consists of 1 min of resting state data across 1772 channels from 106 epilepsy patients (13–62 years old, 58 males, and 48 females), recorded using either surface strip/grid or stereoEEG electrodes, and cleaned of visible artifacts (*Frauscher et al., 2018a*; *Frauscher et al., 2018b*). Neuronal timescales were extracted from PSDs of individual channels, and projected from MNI voxel coordinates onto HCP-MMP1.0 surface parcellation as follows.

For each patient, timescale estimated from each electrode was extrapolated to the rest of the cortex in MNI coordinates using a Gaussian weighting function (confidence mask), $w(r) = e^{-(r^2/\alpha^2)}$, where $r$ is the Euclidean distance between the electrode and a voxel, and $\alpha$ is the distance scaling constant, chosen here such that a voxel 4 mm away has 50% weight (or confidence). Timescale at each voxel is computed as a weighted spatial average of timescales from all electrodes (i) of that patient:

i.e., $\tau_{voxel} = \frac{\sum_i w(r_i)\,\tau_i}{\sum_i w(r_i)}$.

Similarly, each voxel is assigned a confidence rating that is the maximum of weights over all electrodes ($w_{voxel}(r_{min})$, of the closest electrode), i.e., a voxel right under an electrode has a confidence of 1, while a voxel 4 mm away from the closest electrode has a confidence of 0.5, etc.

Timescales for each HCP-MMP parcel were then computed as the confidence-weighted arithmetic mean across all voxels that fall within the boundaries of that parcel. HCP-MMP boundary map is loaded and used for projection using NiBabel (*Brett et al., 2020*). This results in a 180 parcels-by-106 patients timescale matrix. A per-parcel confidence matrix of the same dimensions was computed by taking the maximum confidence over all voxels for each parcel (*Figure 2—figure supplement 1A*). The average cortical timescale map (gradient) is computed by taking the confidence-weighted average at each parcel across all participants. Note that this procedure for locally thresholded and weighted average is different from projection procedures used for the mRNA and T1w/T2w data due to region-constrained and heterogeneous ECoG electrode sites across participants. While coverage is sparse and idiosyncratic in individual participants, it does not vary as a function of age, and when pooling across the entire population, 178 of 180 parcels have at least one patient with an electrode within 4 mm, with the best coverage in sensorimotor, temporal, and frontal regions (*Figure 2—figure supplement 1*).

## T1w/T2w ratio and cortical thickness maps

As a measure of structural cortical hierarchy, we used the ratio between T1- and T2-weighted structural MRI, referred to as T1w/T2w map in main text, or the myelin map (*Burt et al., 2018*; *Glasser and Van Essen, 2011*). Since there is little variation in the myelin map across individuals, we used the group average myelin map of the WU-Minn HCP S1200 release (N = 1096, March 1, 2017 release) provided in HCP-MMP1.0 surface space. For correlation with other variables, we computed the median value per parcel, identical to the procedure for mRNA expression below. Cortical thickness map was similarly generated.

## mRNA expression maps

We used the Allen Human Brain Atlas (AHBA) gene expression dataset (*Hawrylycz et al., 2015*; *Hawrylycz et al., 2012*) that comprised postmortem samples of six donors (one female and five males) that underwent microarray transcriptional profiling. Spatial maps of mRNA expression were available in volumetric 2 mm isotropic MNI space, following improved nonlinear registration and whole-brain prediction using variogram modeling as implemented by *Gryglewski et al., 2018*. We used whole-brain maps available from *Gryglewski et al., 2018* rather than the native sample-wise values in the AHBA database to prevent bias that could occur due to spatial inhomogeneity of the

sampled locations. In total, 18,114 genes were included for analyses that related to the dominant axis of expression across the genome.

We projected the volumetric mRNA expression data onto the HCP-MMP cortical surface using the HCP workbench software (v1.3.1 running on Windows OS 10) with the 'enclosing' method and custom MATLAB code (github.com/rudyvdbrink/surface_projection) (*van den Brink, 2020*). The enclosing method extracts for all vertices on the surface the value from enclosing voxels in the volumetric data. Alternative projection methods such as trilinear 3D linear interpolation of surrounding voxels, or ribbon mapping that constructs a polyhedron from each vertex's neighbors on the surface to compute a weighted mean for the respective vertices, yielded comparable values, but less complete cortical coverage. Moreover, the enclosing method ensured that no transformation of the data (nonlinear or otherwise) occurred during the projection process and thus the original values in the volumetric data were preserved.

Next, for each parcel of the left hemisphere in HCP-MMP, we extracted the median vertex-wise value. We used the median rather than the mean because it reduced the contribution of outliers in expression values within parcels. Vertices that were not enclosed by voxels that contained data in volumetric space were not included in the parcel-wise median. This was the case for 539 vertices (1.81% of total vertices). Linear interpolation across empty vertices prior to computing median parcel-wise values yielded near-identical results ($r = 0.95$ for reconstructed surfaces). Lastly, expression values were mean and variance normalized across parcels to facilitate visualization. Normalization had no effect on spatial correlation between gene expression and other variables since the spatial distribution of gene expression was left unaltered.

## Selection of brain-specific genes

Similar to *Burt et al., 2018*; *Fagerberg et al., 2014*; *Genovese et al., 2016*, N = 2429 brain-specific genes were selected based on the criteria that expression in brain tissues were four times higher than the median expression across all tissue types, using Supplementary Dataset 1 of *Fagerberg et al., 2014*. PC1 result shown in *Figure 3A* is computed from brain-specific genes, though findings are similar when using all genes ($\rho = -0.56$ with timescale map, *Figure 3—figure supplement 1*).

## Spatial statistics

All correlations between spatial maps (timescale, T1w/T2w, gene principal component [PC], and individual gene expressions) were computed using Spearman rank correlation. As noted in *Burt et al., 2020*; *Burt et al., 2018*; *Vos de Wael et al., 2020*, neural variables vary smoothly and continuously across the cortical surface, violating the assumption of independent samples. As a result, when correlating two variables, each with nontrivial SA, the naive p-value is artificially lowered since it is compared against an inappropriate null hypothesis, i.e., randomly distributed or shuffled values across space. Instead, a more appropriate null hypothesis introduces SA-preserving null maps, which destroys any potential correlation between two maps while respecting their SAs. For all spatial correlation analyses, we generated N = 1000 null maps of one variable (timescale map unless otherwise noted), and the test statistic, Spearman correlation ($\rho$), is computed against the other variable of interest to build the null distribution. Two-tailed significance is then computed as the proportion of the null distribution that is less extreme than the empirical correlation value. All regression lines were computed by fitting a linear regression to log-timescale and the structural feature maps.

## SA modeling

To generate SA-preserving null maps, we used Moran Spectral Randomization (MSR) (*Wagner and Dray, 2015*) from the python package BrainSpace (*Vos de Wael et al., 2020*). Details of the algorithm can be found in the above references. Briefly, MSR performs eigendecomposition on a spatial weight matrix of choice, which is taken here to be the inverse average geodesic distance matrix between all pairs of parcels in HCP-MMP1.0. The eigenvectors of the weight matrix are then used to generate randomized null feature maps that preserves the autocorrelation of the empirical map. We used the singleton procedure for null map generation. All significance values reported (*Figures 2D and 3A–C*) were adjusted using the above procedure.

We also compare two other methods of generating null maps: spatial variogram fitting (VF) (*Burt et al., 2020*) and spin permutation (*Alexander-Bloch et al., 2018*). Null maps were generated for timescale using spatial VF, while for spin permutation they were generated for vertex-wise T1w/T2w and gene PC1 maps before parcellation, so as to preserve surface locations of the parcellation itself. All methods perform similarly, producing comparable SA in the null maps, assessed using spatial variogram, as well as null distribution of spatial correlation coefficients between timescale and T1w/T2w (*Figure 2—figure supplement 2*).

## Principal component analysis (PCA) of gene expression

We used scikit-learn (*Pedregosa, 2011*) PCA (sklearn.decomposition.PCA) to identify the dominant axes of gene expression variation across the entire AHBA dataset, as well as for brain-specific genes. PCA was computed on the variance-normalized average gene expression maps, $X$, an N × P matrix where N = 18,114 (or N = 2429 brain-specific) genes, and P = 180 cortical parcels. Briefly, PCA factorizes $X$ such that $X = USV^T$, where $U$ and $V$ are unitary matrices of dimensionality N × N and P × P, respectively. $S$ is the same dimensionality as $X$ and contains non-negative descending eigenvalues on its main diagonal ($\Lambda$). Columns of $V$ are defined as the PCs, and the dominant axis of gene expression is then defined as the first column of V, whose proportion of variance explained in the data is the first element of $\Lambda$ divided by the sum over $\Lambda$. Results for PC1 and PC2-10 are shown in *Figure 3A* and *Figure 3—figure supplement 1*, respectively.

## Comparison of timescale-transcriptomic association with single-cell timescale genes

Single-cell timescale genes were selected based on data from Table S3 of *Tripathy et al., 2017* and Online Table 1 of *Bomkamp et al., 2019*. Using single-cell RNA sequencing data and patch-clamp recordings from transgenic mice cortical neurons, these studies identified genes whose expression significantly correlated with electrophysiological features derived from generalized linear integrate and fire (GLIF) model fits. We selected genes that were significantly correlated with membrane time constant (*tau*), input resistance (*Rin* or *ri*), or capacitance (*Cm* or *cap*) in the referenced data tables, and extracted the level of association between gene expression and those electrophysiological feature (correlation 'DiscCorr' in *Tripathy et al., 2017* and linear coefficient 'beta_gene' in *Bomkamp et al., 2019*).

To compare timescale-gene expression association at the single-cell and macroscale level, we correlated the single-cell associations extracted above with the spatial correlation coefficient (macroscale $\rho$) between ECoG timescale and AHBA microarray expression data for those same genes, restricting to genes with p<0.05 for macroscale correlation (results identical for non-restrictive gene set). Overall association for all genes, as well as split by quintiles of their absolute macroscale correlation coefficient, are shown in *Figure 3D*. Example 'single-cell timescale' genes shown in *Figure 3B and C* are genes showing the highest correlations with those electrophysiology features reported in Table 2 of *Bomkamp et al., 2019*.

## T1w/T2w-removed timescale and gene expression residual maps

To remove anatomical hierarchy as a potential mediating variable in timescale–gene expression relationships, we linearly regress out the T1w/T2w map from the (log) timescale map and individual gene expression maps. T1w/T2w was linearly fit to log-timescale, and the error between T1w/T2w-predicted timescale and empirical timescale was extracted (residual); this identical procedure was applied to every gene expression map to retrieve the gene residuals. SA-preserving null timescale residual maps were similarly created using MSR.

## PLS regression model

Due to multicollinearity in the high-dimensional gene expression dataset (many more genes than parcels), we fit a PLS model to the timescale map with one output dimension (sklearn.cross_decomposition.PLSRegression) to estimate regression coefficient for all genes simultaneously, resulting in N = 18,114 (or N = 2429 brain-specific) PLS weights (*Vértes et al., 2016*; *Whitaker et al., 2016*). To determine significantly associated (or 'enriched') genes, we repeated the above PLS-fitting procedure 1000 times but replaced the empirical timescale map (or residual map) with null timescale

maps (or residual maps) that preserved its SA. Genes whose absolute empirical PLS weight was greater than 95% of its null weight distribution was deemed to be enriched, and submitted for GOEA.

## Gene ontology enrichment analysis

The Gene Ontology (GO) captures hierarchically structured relationships between GO items representing aspects of biological processes (BP), cellular components (CC), or molecular functions (MF). For example, 'synaptic signaling', 'chemical synaptic transmission', and 'glutamatergic synaptic transmission' are GO items with increasing specificity, with smaller subsets of genes associated with each function. Each GO item is annotated with a list of genes that have been linked to that particular process or function. GOEA examines the list of enriched genes from above to identify GO items that are more associated with those genes than expected by chance. We used GOATOOLS (*Klopfenstein et al., 2018*) to perform GOEA programmatically in python.

The list of unranked genes with significant empirical PLS weights was submitted for GOEA as the 'study set', while either the full ABHA list or brain-specific gene list was used as the 'reference set'. The output of GOEA is a list of GO terms with annotated genes that are enriched or purified (i.e., preferentially appearing or missing in the study list, respectively) more often than by chance, determined by Fisher's exact test.

Enrichment ratio is defined as follows: given a reference set with $N$ total genes, and $n$ were found to be significantly associated with timescale (in the study set), for a single GO item with $B$ total genes annotated to it, where $b$ of them overlap with the study set, then. Statistical significance is adjusted for multiple comparisons following Benjamini–Hochberg procedure (false discovery rate q-value reported in *Figure 3F*), and all significant GO items (q < 0.05) are reported in *Figure 3F*, in addition to some example items that did not pass significance threshold. For a detailed exposition, see *Bauer, 2017*. *Figure 3F* shows results using brain-specific genes. The GO items that are significantly associated are similar when using the full gene set, but typically with larger q-values (*Supplementary file 1* and *2*) due to a much larger set of (non-brain-specific) genes. Control analysis was conducted using T1w/T2w, with 1000 similarly generated null maps, instead of timescale.

## Working memory ECoG data and analysis

The CRCNS fcx-2 and fcx-3 datasets include 17 intracranial ECoG recordings in total from epilepsy patients (10 and 7, respectively) performing the same visuospatial working memory task (*Johnson, 2019*; *Johnson, 2018c*; *Johnson et al., 2018a*, *Johnson et al., 2018b*). Subject 3 (s3) from fcx-2 was discarded due to poor data quality upon examination of trial-averaged PSDs (high noise floor near 20 Hz), while s5 and s7 from fcx-3 correspond to s5 and s8 in fcx-2 and were thus combined. Together, data from 14 unique participants (22–50 years old, five females) were analyzed, with variable and overlapping coverage in PC (n = 14), PFC (n = 13), OFC (n = 8), and MTL (n = 9). Each channel was annotated as belonging to one of the above macro regions.

Experimental setup is described in *Johnson, 2019*; *Johnson, 2018c*; *Johnson et al., 2018a*, *Johnson et al., 2018b* in detail. Briefly, following a 1 s pre-trial fixation period (baseline), subjects were instructed to focus on one of two stimulus contexts ('identity' or 'relation' information). Then two shapes were presented in sequence for 200 ms each. After a 900 or 1150 ms jittered precue delay (delay1), the test cue appeared for 800 ms, followed by another post-cue delay period of the same length (delay2). Finally, the response period required participants to perform a 2-alternative forced choice test based on the test cue, which varied based on trial condition. For our analysis, we collapsed across the stimulus context conditions and compared neuronal timescales during the last 900 ms of baseline and delay periods from the epoched data, which were free of visual stimuli, in order to avoid stimulus-related event-related potential effects. Behavioral accuracy for each experimental condition was reported for each participant, and we average across both stimulus context conditions to produce a single working memory accuracy per participant.

Single-trial power spectra were computed for each channel as the squared magnitude of the Hamming-windowed Fourier Transform. We used 900 ms of data in all three periods (pre-trial, delay1, and delay2). Timescales were estimated by applying spectral parameterization as above, and the two delay-period estimates were averaged to produce a single delay period value. For comparison, we computed single-trial theta (3–8 Hz) and high-frequency activity (high gamma

[*Mukamel et al., 2005*], 70–100 Hz) powers as the mean log-power within those frequency bins, as well as spectral exponent ($\chi$). Single-trial timescale difference between delay and baseline was calculated as the difference of the log timescales due to the non-normal distribution of single-trial timescale estimates. All other neural features were computed by subtracting baseline from the delay period.

All neural features were then averaged across channels within the same regions, then trials, for each participant, to produce per-participant region-wise estimates, and finally averaged across all participants for the regional average in *Figure 4B and C*. Two-sided Mann–Whitney U-tests were used to test for significant differences in baseline timescale between pairs of regions (*Figure 4B*). Two-sided Wilcoxon rank-sum tests were used to determine the statistical significance of timescale change in each region (*Figure 4C*), where the null hypothesis was no change between baseline and delay periods (i.e., delay is 100% of baseline). Spearman rank correlation was used to determine the relationship between neural activity (timescale; theta; high-frequency; $\chi$) change and working memory accuracy across participants (*Figure 4D* and *Figure 4—figure supplement 1*).

### Per-subject average cortical timescale across age

Since electrode coverage in the MNI-iEEG dataset is sparse and nonuniform across participants (*Figure 2—figure supplement 1*), simply averaging across parcels within individuals to estimate an average cortical timescale per participant confounds the effect of age with the spatial effect of cortical hierarchy. Therefore, we instead first normalize each parcel by its max value across all participants before averaging within participants, excluding those with fewer than 10 valid parcels (71 of 106 subjects remaining; results hold for a range of threshold values; *Figure 4—figure supplement 2B*). Spearman rank correlation was used to compute the association between age and average cortical timescale.

### Age–timescale association for individual parcels

Each cortical parcel had a variable number of participants with valid timescale estimates above the consistency threshold, so we compute Spearman correlation between age and timescale for each parcel, but including only those with at least five participants (114 of 180 parcels, result holds for a range of threshold values; *Figure 4—figure supplement 2C*). Spatial effect of age-timescale variation is plotted in *Figure 4F*, where parcels that did not meet the threshold criteria are grayed out. Mean age–timescale correlation from individual parcels was significantly negative under one-sample t-test.

### Data and materials' availability

All data analyzed in this manuscript are from open data sources. All code used for all analyses and plots are publicly available on GitHub at https://github.com/rdgao/field-echos (*Gao, 2020*) and https://github.com/rudyvdbrink/surface_projection (*van den Brink, 2020*). See *Tables 1* and *2* for details.

## Acknowledgements

We would like to thank all the creators and curators of the publicly available datasets used in this study, as well as the developers of the open-source software packages, without whom none of this would be possible. We also thank Andrea Chiba, Eran Mukamel, and the Voytek Lab for their feedback and suggestions. We would like to additionally acknowledge the Deutsche Forschungsgemeinschaft (DFG) for their funding to Tobias H. Donner (SFB936/A7).

## Additional information

### Funding

| Funder | Grant reference number | Author |
| --- | --- | --- |
| Natural Sciences and Engineering Research Council of Canada | CGSD3-488052-2016 | Richard Gao |

| | | |
|---|---|---|
| Katzin Prize | | Richard Gao |
| Alexander von Humboldt Foundation | Humboldt Fellowship | Ruud L van den Brink |
| Alexander von Humboldt Foundation | Feodor Lynen Fellowship | Thomas Pfeffer |
| Alfred P. Sloan Foundation | FG-2015-66057 | Bradley Voytek |
| Whitehall Foundation | 2017-12-73 | Bradley Voytek |
| National Science Foundation | BCS-1736028 | Bradley Voytek |
| National Institutes of Health | R01GM134363-01 | Bradley Voytek |
| School of Medicine, UC San Diego | Shiley-Marcos Alzheimer's Disease Research Center | Bradley Voytek |
| Halicioglu Data Science Institute Fellowship | | Bradley Voytek |

The funders had no role in study design, data collection and interpretation, or the decision to submit the work for publication.

### Author contributions

Richard Gao, Conceptualization, Data curation, Software, Formal analysis, Validation, Investigation, Visualization, Methodology, Writing - original draft, Writing - review and editing; Ruud L van den Brink, Conceptualization, Data curation, Software, Formal analysis, Validation, Visualization, Methodology, Writing - review and editing; Thomas Pfeffer, Conceptualization, Resources, Data curation, Writing - review and editing; Bradley Voytek, Conceptualization, Data curation, Software, Formal analysis, Supervision, Funding acquisition, Investigation, Methodology, Writing - original draft, Writing - review and editing

### Author ORCIDs

Richard Gao (ID) https://orcid.org/0000-0001-5916-6433
Ruud L van den Brink (ID) http://orcid.org/0000-0002-3142-7248
Thomas Pfeffer (ID) https://orcid.org/0000-0001-9561-3085
Bradley Voytek (ID) http://orcid.org/0000-0003-1640-2525

### Decision letter and Author response

Decision letter https://doi.org/10.7554/eLife.61277.sa1
Author response https://doi.org/10.7554/eLife.61277.sa2

## Additional files

### Supplementary files

- Supplementary file 1. Significant items from brain-specific GOEA (data for *Figure 3F*).
- Supplementary file 2. Significant items from all-gene GOEA.
- Supplementary file 3. Significant items from all-gene GOEA with T1w/T2w as control.
- Transparent reporting form

### Data availability

All data analyzed in this manuscript are from open data sources:MNI Open iEEG (https://mni-open-ieegatlas.research.mcgill.ca/); Human Connectome Project S1200 Release (https://www.humancon-nectome.org/study/hcp-young-adult/document/1200-subjects-data-release); Neurotycho Anesthesia and Sleep Task (http://neurotycho.org/anesthesia-and-sleep-task); and Whole brain gene expression (http://www.meduniwien.ac.at/neuroimaging/mRNA.html).All code used for all analyses and plots are publicly available on GitHub at https://github.com/rdgao/field-echos (archived at Zenodo http://doi.org/10.5281/zenodo.4362645) and https://github.com/rudyvdbrink/surface_projection (archived at Zenodo http://doi.org/10.5281/zenodo.4352385). See Tables 1 and 2 for details.

The following previously published datasets were used:

| Author(s) | Year | Dataset title | Dataset URL | Database and Identifier |
|---|---|---|---|---|
| Johnson | 2018 | CRCNS fcx-2 | https://crcns.org/data-sets/fcx/fcx-2/about-fcx-2 | CRCNS, 10.6080/K0 VX0DQD |
| Johnson | 2018 | CRCNS fcx-3 | https://crcns.org/data-sets/fcx/fcx-3/about-fcx-3 | CRCNS, 10.6080/K0 6971S9 |

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
