## [Decision Letter]

**Acceptance summary:**

General value assessment: The brain is a hierarchically organized system. Is information in higher brain areas integrated over different time-scales than in lower brain areas? Gao et al. analyze intracranial recordings in humans across a large part of the neocortex using several new analytical techniques. They find that higher brain areas have longer neuronal time-scales. Neuronal time-scales are correlated with specific gene expression patterns, and are correlated with working memory performance.

**Decision letter after peer review:**

Thank you for submitting your article "Neuronal timescales are functionally dynamic and shaped by cortical microarchitecture" for consideration by *eLife*. Your article has been reviewed by four peer reviewers, one of whom is a member of our Board of Reviewing Editors, and the evaluation has been overseen by Laura Colgin as the Senior Editor. The following individual involved in review of your submission has agreed to reveal their identity: Thilo Womelsdorf (Reviewer #4).

The reviewers have discussed the reviews with one another and the Reviewing Editor has drafted this decision to help you prepare a revised submission.

Summary:

Gao et al. analyze how brain-wide timescales of ECoG signals vary across the cortical hierarchy and relate these timescales to several other aspects of structure, behavior and function. They report the following main findings: 1) Timescales increase with the cortical hierarchy. 2) Time-scales, after regressing out the hierarchical T1w/T2w structure variable, correlate significantly with several genes related to synaptic receptors and ion channels. 3) Time-scales increase with working memory task vs. baseline, and predict working memory performance across subjects. 4) Time-scales decrease with aging, in a region-specific way. These findings are a significant advance in comparison to previous work by considering brain-wide hierarchy at a high spatial and temporal resolution and relating them to behaviour and genetics.

All four reviewers agreed the study is of substantial interest. The study was found of high quality.

Essential revisions:

1) Definition and comparison of timescales:

i) The comparison shown in Figure 2 between spiking time-scale and ECOG time-scale might be problematic, in the sense that the spiking time-scales were taken from the Murray et al., 2014 paper where they were quantified with a different technique. A possible solution would be to quantify time-scales in the same manner as Murray, or maybe there is a convincing argument why this is not a problem.

ii) For the non-specialist reader, the concept of neuronal timescales that is central to the paper should be defined more explicitly in the Introduction ('neuronal timescales' appear in paragraph three, while it gets defined in paragraphs one and two).

iii) Fast and slow responding to sensory versus cue related information may reflect a circular definition of timescales.

iv) The Results text says that the aperiodic components is interpreted as time scale but not how the inference is made, i.e. what quantity is interpreted as time scale.

v) It is difficult to keep track of which timescales are referred to when in the text, e.g. the authors start referring to neuronal timescales after having discussed ECOG based time scales and spike timescales. It seems important for cleanly separating the source of the timescale to denote them with a unique label depending on the source data that give rise to them. Why not using a subscript for spike, epiduralECoG, subduralECoG, intracranialLFP,.… ?

2) Timescales and hierarchy:

i) The correlations shown between transcriptomics and timescales need to be carefully considered. While the authors regress out T1w/T2w residuals, these might just be one structural factor that changes with cortical hierarchy and assumes that the underlying relationships are linear. Hence, it is possible that timescales and gene profiles are correlated with structure but that there is no causal relationship between these genes and timescales. In this sense, the correlation of genes with hierarchy might also yield similar genetic profiles. It would be important to show the correlation of hierarchy with genetic profiles, to see whether this looks different from the correlations that are obtained with timescale.

ii) The authors use T1W/T2W as the measure for cortical hierarchy. This is a gradient-based perspective on cortical hierarchy. However, there are other perspectives on hierarchy that are not gradient-based, but are based on anatomical connectivity, e.g. as pursued by Kennedy and Van Essen (Vezoli et al., 2020). This needs to be discussed.

iii) The manuscript addresses two distinct aspects of neuronal timescales: their relationship to local microarchitecture and their dynamics as a function of task or age. Although there is obviously a strong inter-relationship between these two aspects, this deserves a more extensive discussion. For example, in relation with the previous point, if local microstructural properties predict neuronal timescales, why is it that timescale changes during the delay seem to be ubiquitous (or are they)? And why should such changes (that are overall in the same range) correlate with subject performance in the PFC but not in the other areas? How does this relate to the aging observations? Although this discussion is bound to be speculative. It is important in order to strengthen the link between these two independent avenues of the paper, and to enrich the discussion about the functional role of these dynamic changes in neuronal timescales.

iv) The paper does not consider oscillations, which is fine, but the reader is left wondering how oscillations affect these time-scales. Discussion on this aspect would be useful.

4) Comments on figures, statistics and clarification:

i) Are the rho correlation values corrected for the expected value of the surrogate distribution? That is are they significantly overestimated due to the dependent samples issue? In this case it is recommended to report the corrected correlation values, rather than the raw correlation values.

ii) The correlation performed in Figure 4D is a bit unclear. Are the different dots+lines participants, or is this a binned correlation? If it is a binned correlation, does that represent a problem for the correlation analysis?

iii) It would be useful in Figure 1/2 to show some examples of ECOG time-scales related to the actual underlying signals and PSDs, rather than just illustrating the technique on simulated data, so that the validity of the technique can be judged.

iv) In general it would be useful to report carefully the N's and the dataset that is used for each analysis, because it is easy to get lost in what is what as the authors analyze a huge number of datasets.

v) The technique of removing spatial autocorrelations that influence the p-value appears to be sophisticated and well done. If the authors need another validation, one way of doing this would be to use a cross-validation prediction approach where a subset of subjects is used for training and the other subjects are used for testing.

vi) What are "these" limitations in the subsection “Neuronal timescale can be inferred from frequency domain”?

vii) Figure 1E: how is r2=1 when the dots do not fall on the line?

viii) The description of the methods needs be improved. For example, "we can estimate neuronal timescale from the 'characteristic frequency'" which implies a peak in the spectrum. Yet in the next sentence they write that they extract timescale from aperiodic components.

ix) Subsection “Synaptic and ion channel genes shape timescales of neuronal dynamics”: Are these markers also correlated with cell packing density? If so, it's possible that denser neural networks have longer timescales.

x) Relatedly, how strongly inter-correlated are these genetic markers across the cortex? The authors mostly take a mass-univariate approach except for showing gene-PC1 in Figure 3A. There isn't enough information shown to evaluate whether the top PC is suitable, or whether this PC comprises many/all gene contributions or is driven by a small number, etc.

xi) The modeling results seem to be missing. They appear as a schematic in Figure 1 and are mentioned in the Materials and methods section. Was this model actually used somewhere?

xii) In Figure 2B, some T1w/T2w values are above values of 2, which is not standard. Likewise, several outliers can be observed. This might have impacted the estimation of the regression slope. This slope currently matches the one from Burt et al., 2018, although the data point distribution is different.

xiii) Figure 4B is contradicting Figure 2C as the evidenced timescale hierarchy is different (comparing PC, PFC and OFC). Please explain.

xiv) Figure 4B and C, please show actual data points and justify parametric tests.

xv) Figure 4C: how consistent is the increase in delay period timescales across areas within each subject. In other words, is this a general property of the brain, task-related effects resulting in a non-specific adjustment in neuronal timescales or are there regional differences in the reported increase (you might want to exclude the PFC from the analysis to remove task related effects).

xvi) Given the described age-related effect, did the authors check that the different databases they used sampled from subjects with the same age distribution?

xvii) Legend of Figure 1 is not self-explanatory and a lot of the symbols and information plotted in the figures are not explained. Unfortunately, this information is also missing from the Results section.

5) General dense writing style and clarity:

The manuscript is written to be dense yet terse, which makes it harder to read, particularly given the complexity of the analyses. It feels like it was written for a journal with extreme word limitations. The manuscript would be overall improved if the authors would "loosen their belt" and explain the findings and methods in more detail. Figure legends should be more self-explanatory. Quite often, figure detail description and contextual information are missing both from the text and the figures. This also applies to the supplementary figures.

---

## [Author Response]

Essential revisions:1) Definition and comparison of timescales:i) The comparison shown in Figure 2 between spiking time-scale and ECOG time-scale might be problematic, in the sense that the spiking time-scales were taken from the Murray et al., 2014 paper where they were quantified with a different technique. A possible solution would be to quantify time-scales in the same manner as Murray, or maybe there is a convincing argument why this is not a problem.

The spiking timescales taken from Murray et al., 2014, is estimated by fitting an exponential decay function to the spike count autocorrelogram (or autocorrelation function, ACF), and extracting the decay constant. To clarify, this approach measures a mathematically equivalent quantity (decay time constant) as our presented method (though from spiking data in the above case). However, it is problematic when the data has strong and non-stationary oscillatory and long-term variations, as is the case in most neural field potential data, because the oscillations result in ringing in the ACF, impairing precise fits of the decay function, requiring additional components like cosine terms (see Figure 1D inset). Furthermore, it was recently shown that the decay constant obtained via this approach is biased (underestimates) even when the ground-truth model has just a single exponential decay component, due to finite sample size and noise effects (Zeraati et al., 2020).

Alternatively, via the Wiener-Khinchin theorem (Khintchine, 1934), the decay time constant can be equivalently estimated in the frequency domain as the “knee frequency”, as Figure 1E demonstrates (also see, e.g., (Chaudhuri et al., 2017)). This is the approach we take, where oscillatory activity can be more easily accounted for and removed as Gaussian peaks using our spectral parameterization approach (Donogue et al., 2020), and long-term/scale-free components as the 1/f exponent. This is especially important given oscillations with non-stationary frequencies. For example, a broad oscillatory peak in the power spectrum (e.g., ~10Hz in bandwidth like in Figure 1D) represents drifts in the oscillation frequency over time, which is easily accounted for with a single Gaussian, but requires multiple cosine terms to capture well in the autocorrelation. In addition, neural data is dominated by low frequencies due to its power law nature, though we are often interested in capturing the low-power, high-frequency components accurately as well. Fitting the logged PSD (in frequency domain) de-emphasizes the low frequencies, while this is not achievable by fitting the ACF.

In short, the presented method is a more accurate way of estimating the exponential decay constant from data whose autocorrelation displays multiple mixed components. We now emphasize this distinction in the first part of the Results section to further highlight the advantages of the current method.

ii) For the non-specialist reader, the concept of neuronal timescales that is central to the paper should be defined more explicitly in the Introduction ('neuronal timescales' appear in paragraph three, while it gets defined in paragraphs one and two).

We agree, and thank you for this suggestion. In conjunction with another reviewer comment, we now explicitly consider several definitions of “neuronal timescale” and define them in the Introduction (“…and the temporal dynamics of neuronal populations across the cortex are thought to be an intrinsic property (i.e., neuronal timescale) that enables the representation of information over multiple durations in a hierarchically embedded environment”), as well as how our estimate from ECoG relates to previous definitions in the Discussion.

iii) Fast and slow responding to sensory versus cue related information may reflect a circular definition of timescales.

We apologize for the confusion here: the reference to sensory vs. cue-related information (in the first Introduction paragraph) is meant to convey differences in the timescale of stimuli in the natural environment, and an ongoing theory is that different populations in the brain should track these timescales to produce adaptive behavior. This was conflated with neural timescales in an unhelpful way. We have now expanded this section to be more precise:

“Human brain regions are broadly specialized for different aspects of behavior and cognition, and the temporal dynamics of neuronal populations across the cortex are thought to be an intrinsic property (i.e., neuronal timescale) that enables the representation of information over multiple durations in a hierarchically embedded environment (Kiebel et al., 2008)”.

iv) The Results text says that the aperiodic components is interpreted as time scale but not how the inference is made, i.e. what quantity is interpreted as time scale.

Indeed, this was glossed over. We now specify in the first paragraph of the Results section that neuronal timescale is computed from the knee frequency of the aperiodic component, along with mathematical details and reference to prior works, e.g., “From the knee frequency of the aperiodic component, neural timescale (decay constant) can then be computed exactly as τ=12πfk*.”*

v) It is difficult to keep track of which timescales are referred to when in the text, e.g. the authors start referring to neuronal timescales after having discussed ECOG based time scales and spike timescales. It seems important for cleanly separating the source of the timescale to denote them with a unique label depending on the source data that give rise to them. Why not using a subscript for spike, epiduralECoG, subduralECoG, intracranialLFP,.… ?

Thank you for this suggestion. We agree, there are multiple definitions of neuronal timescale, which is especially confusing when spiking timescales are compared with ECoG timescales. Since all main results pertain to timescale estimated from ECoG recordings (aside from the macaque results), we elected not to include a subscript for that quantity to preserve readability of the prose. To reduce confusion, this is now explicitly stated in the Introduction, and subscripts are now included in Figure 2 and the corresponding Results section when comparing spiking and ECoG timescales.

2) Timescales and hierarchy:i) The correlations shown between transcriptomics and timescales need to be carefully considered. While the authors regress out T1w/T2w residuals, these might just be one structural factor that changes with cortical hierarchy and assumes that the underlying relationships are linear. Hence, it is possible that timescales and gene profiles are correlated with structure but that there is no causal relationship between these genes and timescales. In this sense, the correlation of genes with hierarchy might also yield similar genetic profiles. It would be important to show the correlation of hierarchy with genetic profiles, to see whether this looks different from the correlations that are obtained with timescale.

We agree with the reviewer here: in general, it is very difficult to make causal claims regarding the relationship between these co-occurring gradients. Importantly, It has been shown previously that cortical hierarchy (defined by the T1/T2 gradient) is correlated with the expression profile of many individual genes, as well as the overall patterning (top principal components of genes) (Burt et al., 2018). In that study, they perform a similar analysis (GOEA) and identify a number of gene ontologies items that are significantly overrepresented. In fact, as shown in that study, the T1/T2 gradient is more correlated with gene PC1 than timescale.

All this is to say that we acknowledge this limitation wholeheartedly, since there is clear data that the structural gradient is associated with the gene expression gradient, as is the timescale gradient, and the multiple macroscopic gradients are all likely to align to some capacity (e.g., Wang, 2020). This is exactly the motivation to regress out the contribution of T1/T2 from the timescale and gene gradients in the first place, but, as the reviewer points out, regressing out T1/T2 assumes a linear contribution, which does not account for possible nonlinear relationships.

To address this concern, we make three arguments here: first, as a baseline, if timescale provides no (linear) contribution on top of T1T2, then we should not see any significantly associated genes, but this is not the case, implying that timescales has some additional predictive power, which is the case we make in the paper.

Second, from a physiological point of view, gene expression shapes macro and micro-anatomical features, and subsequently, both gene expression and anatomy shape neural dynamics. So in some sense, neural dynamics (specifically, timescale) is downstream to macroanatomy (T1/T2), and we argue precisely that the other micro-anatomical features, such as synaptic receptor and ion channel density, should contribute to shaping neuronal timescale beyond T1/T2. In other words, neural dynamics have to be causally shaped by *some structural property*, and we show that macro-connectivity is not the sole mediating variable between gene expression and neural dynamics.

Third, we interpret that the reviewer would like to see a similar analysis performed for T1/T2 vs. gene maps, and to show that the gene clusters strongly associated with *structural* hierarchy are different from the ones associated with timescales, and we find this to be the case (Supplementary file 3). Specifically, PLS model fit to T1/T2 and gene expression data yields no significant GO terms when the analysis is restricted to brain-specific genes, compared to ones displayed in Figure 3F (and supplementary file 1) showing high specificity to ion channel and synaptic functions. Repeating the analysis with all genes against T1/T2 does yield GO terms that contain ion channel- and synapse-related genes. However, they are much less specific (compared to Supplementary file 2), including a variety of gene clusters associated with general metabolic processes, signalling pathways, and cellular components. This further strengthens the point that removing the contribution of macroanatomy would result in identifying genes that are more specifically associated with neurodynamics. These points are now raised in the Results and Discussion section, and we further stress the non-causal and linear nature of these analyses (Discussion section: “Collinearity and surrogate nature of post-mortem gene expression gradients”).

ii) The authors use T1W/T2W as the measure for cortical hierarchy. This is a gradient-based perspective on cortical hierarchy. However, there are other perspectives on hierarchy that are not gradient-based, but are based on anatomical connectivity, e.g. as pursued by Kennedy and Van Essen (Vezoli et al., 2020). This needs to be discussed.

Thank you for pointing us to this new paper. Our understanding is that (Burt et al., 2018) demonstrated that the T1/T2 gradient is a pragmatic proxy of anatomical hierarchy in human brains where tracing data is difficult to obtain, but closely follows the definition of hierarchy that Kennedy and colleagues have put forward, validating against macaque tracing data (i.e., laminar specificity of feedforward vs. feedback projections). However, it is true that several perspectives exist for the concept of “cortical hierarchy” – all closely related but qualitatively different – as recently reviewed in (Hilgetag and Goulas, 2020). We now specify this in the Introduction (e.g.: “Anatomical connectivity measures based on tract tracing data, such as laminar feedforward vs. feedback projection patterns, have classically defined a hierarchical organization of the cortex (Felleman and Van Essen, 1991; Hilgetag and Goulas, 2020; Vezoli et al., 2020)”) and provide a more detailed treatment of these perspectives in the Discussion section.

iii) The manuscript addresses two distinct aspects of neuronal timescales: their relationship to local microarchitecture and their dynamics as a function of task or age. Although there is obviously a strong inter-relationship between these two aspects, this deserves a more extensive discussion. For example, in relation with the previous point, if local microstructural properties predict neuronal timescales, why is it that timescale changes during the delay seem to be ubiquitous (or are they)? And why should such changes (that are overall in the same range) correlate with subject performance in the PFC but not in the other areas? How does this relate to the aging observations? Although this discussion is bound to be speculative. It is important in order to strengthen the link between these two independent avenues of the paper, and to enrich the discussion about the functional role of these dynamic changes in neuronal timescales.

We agree, and we believe that the integration of these two aspects is a fascinating avenue of future research. In brief, neuronal dynamics is first and foremost constrained by local and long-range structural properties, including connectivity and cell-type variations, and this has been previously shown in many different ways, both experimentally and through computational modeling. However, these structural properties only experience changes in the (relatively) longer term (e.g., development, long-term plasticity), and a brain where dynamical properties are fixed relative to its structural properties would not be very adaptive (or useful) in the short term.

In contrast, we observe relatively broad changes in timescales over both short and long terms, but area-specific correlation with behavior. The first point is one we intended to emphasize and provide initial evidence for in this manuscript, while providing suggestive interpretations for the second point. Our interpretation is that structural properties define a range for neural dynamic properties (specifically, timescale), while task requirements, input statistics, short term synaptic plasticity, and neuromodulation may broadly shift cortical timescale within this range, but only computations within the area of relevance (i.e., PFC) is indicative of task performance. Following the reviewer’s suggestion, we now provide a more detailed interpretation and some speculation in the Discussion section.

iv) The paper does not consider oscillations, which is fine, but the reader is left wondering how oscillations affect these time-scales. Discussion on this aspect would be useful.

Thank you for this suggestion. We agree – while oscillations are ignored on purpose here when estimating neuronal timescales, oscillatory parameters and aperiodic timescale likely co-evolve as a function of the underlying synaptic, neuronal, and network properties (e.g., time delay between interacting local excitatory and inhibitory populations), and deserve an extended discussion. In brief, we interpret timescale and oscillations to be related dynamical quantities, similar to the decay constant and characteristic (or resonant) frequency in a damped harmonic oscillator, which arise depending on the input and operating regime of the oscillator. Supporting this, a recent study reports an anterior-to-posterior gradient in oscillatory frequency across the human cortex using MEG data, and is also correlated with cortical hierarchy (Mahjoory et al., 2020). We now include these points in the Discussion section.

4) Comments on figures, statistics and clarification:i) Are the rho correlation values corrected for the expected value of the surrogate distribution? That is are they significantly overestimated due to the dependent samples issue? In this case it is recommended to report the corrected correlation values, rather than the raw correlation values.

As the reviewer points out, these samples have non-trivial spatial autocorrelation, and must be accounted for when reporting statistical significance. To do this, all reported p-values are the “corrected p-values”, which are computed by comparing the raw correlation against a surrogate distribution of correlation values using permutation procedures that shuffle the data but preserve spatial autocorrelation (e.g., see Figure 2—figure supplement 2 for surrogate distributions). Consistent with existing literature (e.g., (Burt et al., 2020; Vos de Wael et al., 2020)), all reported rho values are “raw” in the sense that it is the direct Spearman correlation computed from a pair of vectors (e.g., timescale and T1/T2 over space), but the p-values are corrected.

ii) The correlation performed in Figure 4D is a bit unclear. Are the different dots+lines participants, or is this a binned correlation? If it is a binned correlation, does that represent a problem for the correlation analysis?

Each dot is a participant, and the error bars represent the SEM around that mean across all PFC electrodes for that participant. The figure legend for 4D is now updated to reflect this, and we apologize for being unclear here.

iii) It would be useful in Figure 1/2 to show some examples of ECOG time-scales related to the actual underlying signals and PSDs, rather than just illustrating the technique on simulated data, so that the validity of the technique can be judged.

Thank you for this suggestion, this would indeed improve the clarity of the manuscript. Figure 2 now includes example time-series and power spectra from representative electrodes along the cortical hierarchy from both the human and macaque datasets.

iv) In general it would be useful to report carefully the N's and the dataset that is used for each analysis, because it is easy to get lost in what is what as the authors analyze a huge number of datasets.

Agreed – the N’s and details of the dataset are now included in the figure legends, and summarized in Table 1.

v) The technique of removing spatial autocorrelations that influence the p-value appears to be sophisticated and well done. If the authors need another validation, one way of doing this would be to use a cross-validation prediction approach where a subset of subjects is used for training and the other subjects are used for testing.

Thank you for the kind words and the following suggestion. We agree that cross-validation with held-out participants would in general be a good approach. In this particular case, it is made more difficult by the fact that participants have heterogeneous electrode coverage, so a naive k-fold or leave-one-out approach would result in many empty cortical parcels, and thus require sophisticated grouping strategies to ensure complete coverage when averaging the training and validation sets.

vi) What are "these" limitations in the subsection “Neuronal timescale can be inferred from frequency domain”?

Original: “To overcome these limitations, we develop a novel computational method for inferring the timescale of neuronal transmembrane current fluctuations from human intracranial electrocorticography (ECoG) recordings (Figure 1A, box).”

New: “To relate whole-cortex transcriptomic and anatomical data with neuronal dynamic timescales in humans, we develop a novel computational method for inferring the timescale of neuronal transmembrane current fluctuations from intracranial electrocorticography (ECoG) recordings (Figure 1A, box).”

vii) Figure 1E: how is r2=1 when the dots do not fall on the line?

The correlation value is reported up to 3 decimal places, which was rounded up to 1. There is an error here, however, as we intended to report Pearson correlation (r), not r^2^. This is now reflected in the figure and legend.

viii) The description of the methods needs be improved. For example, "we can estimate neuronal timescale from the 'characteristic frequency'" which implies a peak in the spectrum. Yet in the next sentence they write that they extract timescale from aperiodic components.

We apologize for the confusion here. “Characteristic frequency” is terminology borrowed from the filter design literature that corresponds to the roll-off frequency of, for example, a low-pass filter. However, it introduces confusion in this context, so we have replaced it with “knee frequency” and expanded on the description of the timescale estimation procedure in the first paragraph of the Results section.

ix) Subsection “Synaptic and ion channel genes shape timescales of neuronal dynamics”: Are these markers also correlated with cell packing density? If so, it's possible that denser neural networks have longer timescales.

This is a very interesting question: previous theoretical works have demonstrated many network properties (including population size) to be important for creating long timescale dynamics around attractor states, especially in nonlinear and strongly recurrent networks, reviewed in, e.g., (Barak and Tsodyks, 2014; Huang and Doiron, 2017). However, this relationship is difficult to test directly, especially in the human cortex, due to the lack of detailed interareal neuronal count measurements. As such, it’s unclear whether there is a similarly graded variation in cell number from sensory to association cortices. A comprehensive review of mouse data suggests that sensory (especially visual) areas have the highest neuronal density (Keller et al., 2018), though the decreased count in higher order frontal regions may not apply to primate brains. It is possible to use non-invasive proxies to infer cell density in the human cortex. For example, gray matter volume has previously been used as a proxy for neuronal density, and correlates with fMRI timescales across individuals (Watanabe et al., 2019). However, it was shown to be unrelated to direct histological measurements of cell count within the temporal lobe when compared to resected tissue (Eriksson et al., 2009), and cell-type specific contributions inferred from gene expression analysis (using the same ABHA data) show a complex relationship between cell density and grey matter thickness (Shin et al., 2018).

Nevertheless, following this suggestion, we use cortical thickness from the HCP dataset as a surrogate for human cortical cell packing density. We see that thicker regions do have longer timescales (Figure 2—figure supplement 3, left), though this relationship is weaker than that of timescale vs. T1T2. Cortical thickness is also negatively correlated with T1T2 (Figure 2—figure supplement 3, right), furthering the point of multiple overlapping large-scale cortical gradients. As such, we cannot eliminate the possibility that denser networks directly contribute to having longer timescales (at least partially). However, we do not claim that cortical myelination (measured via T1T2) is the only contributor to neuronal timescale, but that timescale is a dynamic-readout of the abstract quantity of cortical hierarchy, which may be defined by gene expression and further shapes macroanatomical measures such as cortical thickness, cell density, and GM myelination. (Burt et al., 2018)(Figure 6) showed that T1T2 was a better predictor of gene expression gradients than cortical thickness, which we take to mean that T1T2 is a more representative measure of cortical hierarchy than cortical thickness. All these interpretations are consistent with each other, and we now address this in the Discussion.

x) Relatedly, how strongly inter-correlated are these genetic markers across the cortex? The authors mostly take a mass-univariate approach except for showing gene-PC1 in Figure 3A. There isn't enough information shown to evaluate whether the top PC is suitable, or whether this PC comprises many/all gene contributions or is driven by a small number, etc.

In general, these gene expression profiles are strongly correlated with each other and have topographies closely related to distinct cortical macroregions, as previous works have shown (e.g., (Burt et al., 2018; Hawrylycz et al., 2012)). As shown in Figure 2—figure supplement 2, the top principal component accounts for nearly 40% of the variance, while each of the first 5 PCs account for around 10% or more, and quickly approach 0 thereafter, implying a low-dimensional structure in the transcriptomic data. In the same figure, we note that only the 1st gene PC is significantly correlated with the timescale gradient after accounting for spatial autocorrelation. To address the reviewer’s question regarding whether many genes or a handful of outliers contribute to PC1, we also plot the distribution of weights for the first 5 PCs (Author response image 1), and observe that the extremes of the data (top and bottom ticks) are well within the main body of the distributions. This is especially true for PC1 where the weights are more uniformly distributed, hence receiving contributions from many genes.

In addition, we note that the partial least squares (PLS) model we use for the gene ontology enrichment analysis is a multivariate approach, where weights for each gene are fit simultaneously and account for multicollinearity. Genes with large weights are then submitted to GOEA, where distinct functional gene clusters (i.e., gene ontology items) can be found. If it is the case that only a couple of genes are strongly associated with timescale, then we would not expect to see any significant gene clusters to be found. In contrast, Figure 3E and F show that many genes – and more importantly, genes belonging to the same functional groups (e.g., ion channel complex) – are associated with the cortical timescale gradient.

xi) The modeling results seem to be missing. They appear as a schematic in Figure 1 and are mentioned in the Materials and methods section. Was this model actually used somewhere?

We apologize for the confusion – ”model” is used to mean the spectral parameterization model with aperiodic and periodic components, like “model” in GLM or linear regression model. In other words, it’s a descriptive/statistical model, not a mechanistic one. The legend in Figure 1D thus refers to the fitted spectrum as the “full model fit” (but we’ve now removed “model” there to avoid confusion). There is no formal computational modeling of a circuit using, e.g., LIF networks, and all simulated time series data are referred to as simulations to avoid confusion.

xii) In Figure 2B, some T1w/T2w values are above values of 2, which is not standard. Likewise, several outliers can be observed. This might have impacted the estimation of the regression slope. This slope currently matches the one from Burt et al., 2018, although the data point distribution is different.

Figure 2B plots the standardized (z-scored) T1/T2 values on the x-axis, not their raw values (which range between ~1.07 and ~1.73), in order to be consistent with the topography in 2A and similar presentations of the gene expression data. We apologize for this confusion, and have emphasized this point accordingly in the figure and legend. Taking the reviewer’s suggestion, we also performed the correlation after removing all T1/T2 values outside of 2 standard deviations (Author response image 2), and we observed that the reported association is now even stronger (though we keep with the original data in Figure 2).

**Author response image 2. respfig2:** 

xiii) Figure 4B is contradicting Figure 2C as the evidenced timescale hierarchy is different (comparing PC, PFC and OFC). Please explain.

In terms of cortical hierarchy, we expect timescales to increase from parietal regions (PC), to PFC, to OFC. This is observed in macaques in Murray et al., 2014 (which is plotted on the x-axis in Figure 2C), though PFC and OFC timescales are very similar. While we are hesitant to overinterpret, comparative anatomical studies across primates offer some insights regarding these differences. Compared to other primates, including macaques, the human OFC is larger and more specialized compared to surrounding frontal areas, and to other primates (Semendeferi et al., 2001). Since the evolutionary "distance" between PFC and OFC in humans is much greater than that in macaques, it could explain their relatively large difference in Figure 4B but not 2C (now 2F). In Figure 4C, we see that while this relationship is roughly preserved in humans, pairwise t-tests show that PC, PFC, and OFC are not statistically different from each other, and only MTL timescales are significantly longer than PC and PFC. Therefore, we cannot make a definite conclusion about the hierarchical ordering of these 3 regions from timescale alone, since numerical results appear to be statistical fluctuations. However, to further examine this in the larger MNI-iEEG human ECoG database, we grouped the 180 Glasser parcels into 21 macro-regions with broader regional labels (as discussed in Supplementary Materials of (Glasser et al., 2016)). Sorting by their average timescale (mean±sem), we see that the expected ranking is reproduced, where timescale increases from parietal regions, to dlPFC/mPFC, to OFC, and to MTL. We now include this as Figure 2—figure supplement 1G.

xiv) Figure 4B and C, please show actual data points and justify parametric tests.

Thank you for this suggestion – Figure 4B and C now show smaller dots for individual subjects, and we now use non-parametric tests (Mann-Whitney U test and Wilcoxon ranked signed rank test for 4B and C respectively), with nearly identical conclusions (with the exception that parietal cortex baseline timescale is now significantly faster than OFC in 4C). Figures and legends are updated accordingly.

xv) Figure 4C: how consistent is the increase in delay period timescales across areas within each subject. In other words, is this a general property of the brain, task-related effects resulting in a non-specific adjustment in neuronal timescales or are there regional differences in the reported increase (you might want to exclude the PFC from the analysis to remove task related effects).

Following an earlier comment (Point 4-ii), we now plot the per-area timescale change for each participant as individual dots in Figure 4C (each small dot is a participant, averaged over all electrodes in that region; large circle is mean across participants), and observe that the overall increase in timescale is very consistent across brain regions and subjects (even for non-task correlated regions).

Furthermore, pairwise Mann-Whitney U tests fail to detect significant differences in timescale change between any two regions, suggesting that the increase in timescale is general, at least within the association regions where electrode coverage exists in this dataset.

xvi) Given the described age-related effect, did the authors check that the different databases they used sampled from subjects with the same age distribution?

To clarify, we used two different human ECoG databases in the study, and the timescale-age analysis (Figure 4E and F) was done exclusively on the MNI-iEEG database due to the large participant pool with a wide age range (N = 106, ages 13-65, 33.40±10.63 years old, distribution – Author response image 3).

**Author response image 3. respfig3:** 

Structural and gene expression data used for participant-average comparison were from cohorts with similar age distributions (HCP S1200: N = 1096, age: 22-36+ (details restricted due to identifiability); Allen Human Brain Atlas: N = 6, age: 42.5±12.2; now summarized in Table 1). We further ensure that the average timescale map is not biased by age-related implant locations (Figure 2—figure supplement 1).For the working memory analysis in Figure 4A-D, we used the fcx-2 and fcx-3 datasets from CRCNS (N = 14, age 30.9±7.8; one person age 50, everyone else between 22-42). This cohort has a much narrower age range and restricted electrode coverage, and is hence excluded from age-related analysis.

xvii) Legend of Figure 1 is not self-explanatory and a lot of the symbols and information plotted in the figures are not explained. Unfortunately, this information is also missing from the Results section.

Following this suggestion, a more detailed description, especially of Figure 1D, is now provided in the figure legend (“(D) example macaque ECoG power spectral density (PSD) showing the aperiodic component fit (red dashed), and the “knee frequency” at which power drops off (fk, red circle; insets: time series and ACF).”) and first section of the Results section.

5) General dense writing style and clarity:The manuscript is written to be dense yet terse, which makes it harder to read, particularly given the complexity of the analyses. It feels like it was written for a journal with extreme word limitations. The manuscript would be overall improved if the authors would "loosen their belt" and explain the findings and methods in more detail. Figure legends should be more self-explanatory. Quite often, figure detail description and contextual information are missing both from the text and the figures. This also applies to the supplementary figures.

We agree with the reviewer here – we have now expanded on several of the places outlined here, including figure legends and all sections of the main text, especially pertaining to the method, as several other comments have suggested. Much more detailed discussions on results, assumptions, and limitations of the analyses are now included as well.

**References:**

Barak, O., and Tsodyks, M. (2014). Working models of working memory. Current Opinion in Neurobiology, 25, 20–24.

Eriksson, S. H., Free, S. L., Thom, M., Symms, M. R., Martinian, L., Duncan, J. S., and Sisodiya, S. M. (2009). Quantitative grey matter histological measures do not correlate with grey matter probability values from in vivo MRI in the temporal lobe. Journal of Neuroscience Methods, 181(1), 111–118.

Keller, D., Erö, C., and Markram, H. (2018). Cell Densities in the Mouse Brain: A Systematic Review. Frontiers in Neuroanatomy, 12, 83.

Semendeferi, K., Armstrong, E., Schleicher, A., Zilles, K., and Van Hoesen, G. W. (2001). Prefrontal cortex in humans and apes: a comparative study of area 10. American Journal of Physical Anthropology, 114(3), 224–241.

Shin, J., French, L., Xu, T., Leonard, G., Perron, M., Pike, G. B., Richer, L., Veillette, S., Pausova, Z., and Paus, T. (2018). Cell-Specific Gene-Expression Profiles and Cortical Thickness in the Human Brain. Cerebral Cortex , 28(9), 3267–3277.